# $G_{i/o}$ protein-coupled receptor inhibition of beta-cell electrical excitability and insulin secretion depends on Na$^+$/K$^+$ ATPase activation

Matthew T. Dickerson[1], Prasanna K. Dadi [1], Karolina E. Zaborska[1], Arya Y. Nakhe[1], Charles M. Schaub[1], Jordyn R. Dobson[1], Nicole M. Wright[1], Joshua C. Lynch[1], Claire F. Scott [1], Logan D. Robinson[1] & David A. Jacobson [1]✉

$G_{i/o}$-coupled somatostatin or α2-adrenergic receptor activation stimulated β-cell NKA activity, resulting in islet Ca$^{2+}$ fluctuations. Furthermore, intra-islet paracrine activation of β-cell $G_{i/o}$-GPCRs and NKAs by δ-cell somatostatin secretion slowed Ca$^{2+}$ oscillations, which decreased insulin secretion. β-cell membrane potential hyperpolarization resulting from $G_{i/o}$-GPCR activation was dependent on NKA phosphorylation by Src tyrosine kinases. Whereas, β-cell NKA function was inhibited by cAMP-dependent PKA activity. These data reveal that NKA-mediated β-cell membrane potential hyperpolarization is the primary and conserved mechanism for $G_{i/o}$-GPCR control of electrical excitability, Ca$^{2+}$ handling, and insulin secretion.

Pancreatic β-cell glucose-stimulated insulin secretion (GSIS) is essential for maintenance of euglycemia[1,2], and as Ca$^{2+}$ entry is required for GSIS, mechanisms that control β-cell Ca$^{2+}$ handling are critical regulators of blood glucose homeostasis[3–7]. It was discovered more than 50 years ago that $G_{i/o}$ protein-coupled receptors ($G_{i/o}$-GPCRs) play a critical role in limiting insulin secretion in-part by decreasing β-cell electrical excitability and subsequent Ca$^{2+}$ influx[8–10]. However, the exact mechanism(s) of $G_{i/o}$-GPCR control of β-cell electrical excitability, Ca$^{2+}$ handling, and insulin secretion remain poorly understood.

β-cells express numerous $G_{i/o}$-GPCRs such as somatostatin receptors (SSTRs), α2A-adrenergic receptors (ADRs), and D2-like dopamine receptors (DRDs)[11–16]. As a consequence, treatment of islets with $G_{i/o}$-GPCR ligands (i.e., somatostatin (SST), adrenaline, or dopamine) activates hyperpolarizing currents and reduces intracellular cAMP ([cAMP]$_i$) levels, which results in decreased intracellular Ca$^{2+}$ ([Ca$^{2+}$]$_i$) and diminished insulin secretion[16–21]. Insulin secretion is inhibited by $G_{i/o}$-GPCRs, therefore, these signals are critical for preventing excessive insulin secretion under hypoglycemic as well as stimulatory conditions[22]. Indeed, loss of intact α2-ADR signaling leads to a drop in blood glucose levels under fasting and fed conditions due to

elevated insulin secretion[23], while α-ADR agonists attenuate GSIS[24,25]. Intra-islet communication is also mediated via $G_{i/o}$ signaling (i.e., SST secreted by δ-cells, dopamine secreted by β- and α-cells)[26–30], which tunes β-cell Ca$^{2+}$ handling and insulin secretion. Thus, inhibition of islet $G_{i/o}$-GPCRs with pertussis toxin, also known as islet activating protein[31], significantly stimulates hormone secretion, highlighting the importance of $G_{i/o}$-GPCRs in regulating islet function. As numerous $G_{i/o}$-GPCRs control physiological β-cell function, perturbations in these pathways impair β-cell GSIS and are in some instances associated with increased risk of developing diabetes[14,32–34]. For example, glucose-stimulated SST secretion is blunted during the pathogenesis of type 2 diabetes (T2D), which diminishes SSTR-mediated control of β-cell function[33]. Moreover, polymorphisms that increase α2A-ADR expression result in increased risk of developing T2D due to suppression of GSIS[14,32,34]. Taken together, these findings strongly suggest that $G_{i/o}$ signaling plays a key role in regulating β-cell Ca$^{2+}$ handling and insulin secretion; however, the underlying mechanism has not been conclusively identified for more than half a century.

β-cell $V_m$ hyperpolarization is predominantly mediated by K$^+$ efflux; thus, it has been generally accepted that $G_{i/o}$-GPCR signaling

[1]Molecular Physiology and Biophysics Department, Vanderbilt University, 7425B MRB IV, 2213 Garland Ave., Nashville, TN, USA.
✉e-mail: david.a.jacobson@vanderbilt.edu

activates an outward K$^+$ conductance. ATP-sensitive K$^+$ (K$_{ATP}$) channels can be ruled out as the source, because G$_{i/o}$-GPCR activation induces β-cell $V_m$ hyperpolarization in the presence of sulfonylureas and in K$_{ATP}$ channel-deficient islets[21,35]. As is the case in numerous other tissues, G$_{i/o}$ signaling-induced β-cell $V_m$ hyperpolarization has been widely ascribed to activation of G protein-gated inwardly-rectifying K$^+$ (GIRK) channels[16,36]. RNA sequencing studies show that both mouse and human β-cells express low levels of GIRK channel transcripts (predominantly *KCNJ6*, the gene encoding GIRK2)[11,12,16], and immunofluorescent staining of mouse pancreatic sections confirms the expression of GIRK channel proteins in β-cells[36]. However, while G$_{i/o}$ signaling activates robust inwardly-rectifying K$^+$ currents in other cell types where GIRK channels are expressed[37], currents with GIRK-like characteristics have not been reproducibly observed in primary β-cells. Moreover, there are conflicting reports detailing the effect of GIRK channel inhibitors on β-cell electrical activity and Ca$^{2+}$ handling. For example, one study determined that pharmacological GIRK channel inhibition blocks adrenaline-induced $V_m$ hyperpolarization in rat β-cells[36], but another manuscript found that GIRK channel inhibition does not prevent adrenaline-induced $V_m$ hyperpolarization in mouse β-cells[19]. Furthermore, treatment with a wide range of other K$^+$ channel blockers failed to inhibit adrenaline-induced β-cell $V_m$ hyperpolarization. These observations suggest that G$_{i/o}$ signaling-induced β-cell $V_m$ hyperpolarization is not mediated by GIRKs or other K$^+$ channels.

Electrogenic Na$^+$/K$^+$ ATPases (NKAs) can also be activated by G$_{i/o}$-GPCR signaling leading to $V_m$ hyperpolarization[35,38,39]. NKA α1 pore-forming subunits (encoded by *ATP1A1*) are highly expressed in mouse and human β-cells[11-13] where they preserve steep ionic gradients that are essential for setting and maintaining $V_m$. Active β-cell NKAs generate a net outward cationic flux by extruding three intracellular Na$^+$ ions in exchange for two extracellular K$^+$ ions, resulting in $V_m$ hyperpolarization[35,38,39]. Inhibition of β-cell NKAs with ouabain leads to $V_m$ depolarization and enhanced insulin secretion[40,41], illustrating the key role that NKAs serve in preventing excessive insulin secretion. Importantly, NKA activity is regulated by several β-cell protein kinases that are influenced by G$_{i/o}$-GPCR signaling. For example, protein kinase A (PKA), which limits NKA activity, is inhibited by SSTR signaling in a cAMP-dependent manner; whereas, Src tyrosine kinases (STKs) augment NKA function and are activated by SSTR signaling[38,42-46]. Thus, G$_{i/o}$ signaling is predicted to influence β-cell Ca$^{2+}$ handling through changes in NKA α1 subunit phosphorylation.

Here we show that NKA-mediated β-cell $V_m$ hyperpolarization is a principal mechanism for G$_{i/o}$-GPCR control of β-cell Ca$^{2+}$ handling and insulin secretion. Activation of β-cell G$_{i/o}$ signaling generated ouabain- and K$^+$-sensitive outward currents as well as [Ca$^{2+}$]$_i$ oscillations independently of K$_{ATP}$ in both mouse and human β-cells. Inhibition of SST secretion from δ-cells increased islet [Ca$^{2+}$]$_i$, accelerated islet [Ca$^{2+}$]$_i$ oscillations, and enhanced GSIS, demonstrating the critical role of δ-cell paracrine signaling in tuning β-cell Ca$^{2+}$ handling and insulin secretion, likely through NKA activation. cAMP-dependent PKA activation decreased β-cell NKA function, whereas stimulation of tyrosine kinases (STKs, insulin receptors) initiated islet [Ca$^{2+}$]$_i$ oscillations. These results strongly suggest that phosphorylation by PKA and tyrosine kinases serve as crucial counter-regulatory mechanisms for modulation of β-cell NKA activity. Therefore, these findings illuminate a conserved mechanism for G$_{i/o}$-GPCR control of β-cell NKAs, which plays a key role in regulating β-cell electrical excitability, Ca$^{2+}$ handling, and insulin secretion.

## Results

### SSTR signaling activates GIRK channel-independent outward currents

It is generally accepted that β-cell G$_{i/o}$ signaling activates hyperpolarizing GIRK channels[16,36]; however, measurement of β-cell GIRK currents has proven difficult utilizing traditional voltage-clamp recording techniques. Thus, we employed a modified recording paradigm to elicit quantifiable SST-induced β-cell currents. SST-mediated changes in β-cell $V_m$ were monitored in intact islets and whole-cell currents were recorded in response to voltage ramps before and after each treatment[47-49]. β-cell currents displayed minimal trace-to-trace variability in response to repeated voltage ramps indicating stable voltage-clamping; however, to account for small current distortions arising from electrical activity in neighboring β-cells within an intact islet[50], all currents were calculated from the median of at least 10 consecutive traces. In wild type (WT) mouse islets maximally stimulated with 20 mM glucose and 1 mM tolbutamide[21], 200 nM SST induced outward β-cell currents with little rectification (current amplitude at −50 mV: 15.7 ± 1.9 pA; Fig. 1A–C; $P < 0.0001$); SST also hyperpolarized $V_m$ (−30.2 ± 3.5 mV; Fig. 1A, D; $P < 0.0001$).

As transcriptome studies indicate that *Kcnj6* is the most abundant islet GIRK channel transcript[11-13], GIRK2 deficient (GIRK2 KO$^{Panc}$) mouse islets were utilized to determine the contribution of these channels to SST-induced β-cell currents. As observed in WT β-cells, 200 nM SST elicited outward non-rectifying currents in β-cells without GIRK2 channels (current amplitude at −50 mV: 17.6 ± 2.0pA; Fig. 1E, F; $P < 0.0001$) and hyperpolarized $V_m$ (−23.2 ± 2.6 mV; Fig. 1G and Supplementary Fig. 1; $P < 0.01$). Tolbutamide-stimulated mouse islets displayed transient glucose-mediated (20 mM) [Ca$^{2+}$]$_i$ decreases, after which [Ca$^{2+}$]$_i$ stabilized at an elevated level (Fig. 1H). Under these conditions, 200 nM SST induced [Ca$^{2+}$]$_i$ oscillations in 95.3 ± 2.4% of GIRK2 KO$^{Panc}$ islets, which was indistinguishable from WT islets (94.7 ± 4.8%; Fig. 1I–K), and reduced GIRK2 KO$^{Panc}$ islet [Ca$^{2+}$]$_i$ plateau fraction by 32.6 ± 4.2% compared to before treatment (Fig. 1J, L; $P < 0.01$). GIRK channels were also pharmacologically inhibited in WT islets with 200 nM tertiapin-Q (TPQ) to confirm GIRK channel-independent SST-induced β-cell currents[16,19]. In the presence of TPQ, 200 nM SST elicited outward β-cell currents with minimal inward rectification (max at −50 mV: 12.4 ± 1.8pA; Fig. 1M–O; $P < 0.0001$) and hyperpolarized $V_m$ (−25.2 ± 6.7 mV; Fig. 1M, P; $P < 0.001$). Although β-cell currents trended lower in the presence of TPQ, SST-induced β-cell currents were not significantly decreased when GIRK channels were inhibited. Furthermore, 76.8 ± 10.5% of islets displayed SST-induced [Ca$^{2+}$]$_i$ oscillations following treatment with 200 nM TPQ, which was not significantly different than with SST alone (92.8 ± 7.3%; Fig. 1Q, R). However, TPQ treatment after SST did modestly increase islet [Ca$^{2+}$]$_i$ plateau fraction by 13.7 ± 4.8% (Fig. 1Q, S; $P < 0.05$). These results demonstrate that while GIRK channels likely play a role in G$_{i/o}$-GPCR regulation of β-cell electrical activity, SST-induced β-cell hyperpolarization is largely independent of GIRK channel activity.

### SSTR signaling induces islet [Ca$^{2+}$]$_i$ oscillations by stimulating β-cell NKA activity

SST stimulated outward currents below the equilibrium potential of K$^+$, which suggests that SST-induced β-cell currents are not mediated by K$^+$ channels (Fig. 1C, F, O). This instead indicates that SST-induced currents are likely due to efflux of another cation such as Na$^+$, which requires movement against an ion concentration gradient through energy-dependent ion pumps. Therefore, we investigated whether G$_{i/o}$ signaling decreases β-cell [Ca$^{2+}$]$_i$ by facilitating a net outflow of positive charge through electrogenic NKAs. In islets undergoing SST-mediated [Ca$^{2+}$]$_i$ oscillations treatment with 150 μM ouabain (Oua) or removal of extracellular K$^+$ (Fig. 2A–C) resulted in sustained elevation of [Ca$^{2+}$]$_i$ in almost all islets (97.1 ± 2.9% [$P < 0.001$] and 96.1 ± 1.9% [$P < 0.0001$] of islets respectively). Oua treatment subsequent to removal of extracellular K$^+$ had no additional effect on islet [Ca$^{2+}$]$_i$, suggesting that islet NKAs are completely inhibited under these conditions (Fig. 2B). Treatment with 200 nM SST significantly decreased islet [Ca$^{2+}$]$_i$ plateau fraction with (54.8 ± 5.8% decrease; $P < 0.0001$) and without tolbutamide-mediated K$_{ATP}$ inhibition (76.4 ± 6.4% decrease; Fig. 2A, B, and D; $P < 0.0001$). Importantly, K$_{ATP}$ activation with 125 μM diazoxide

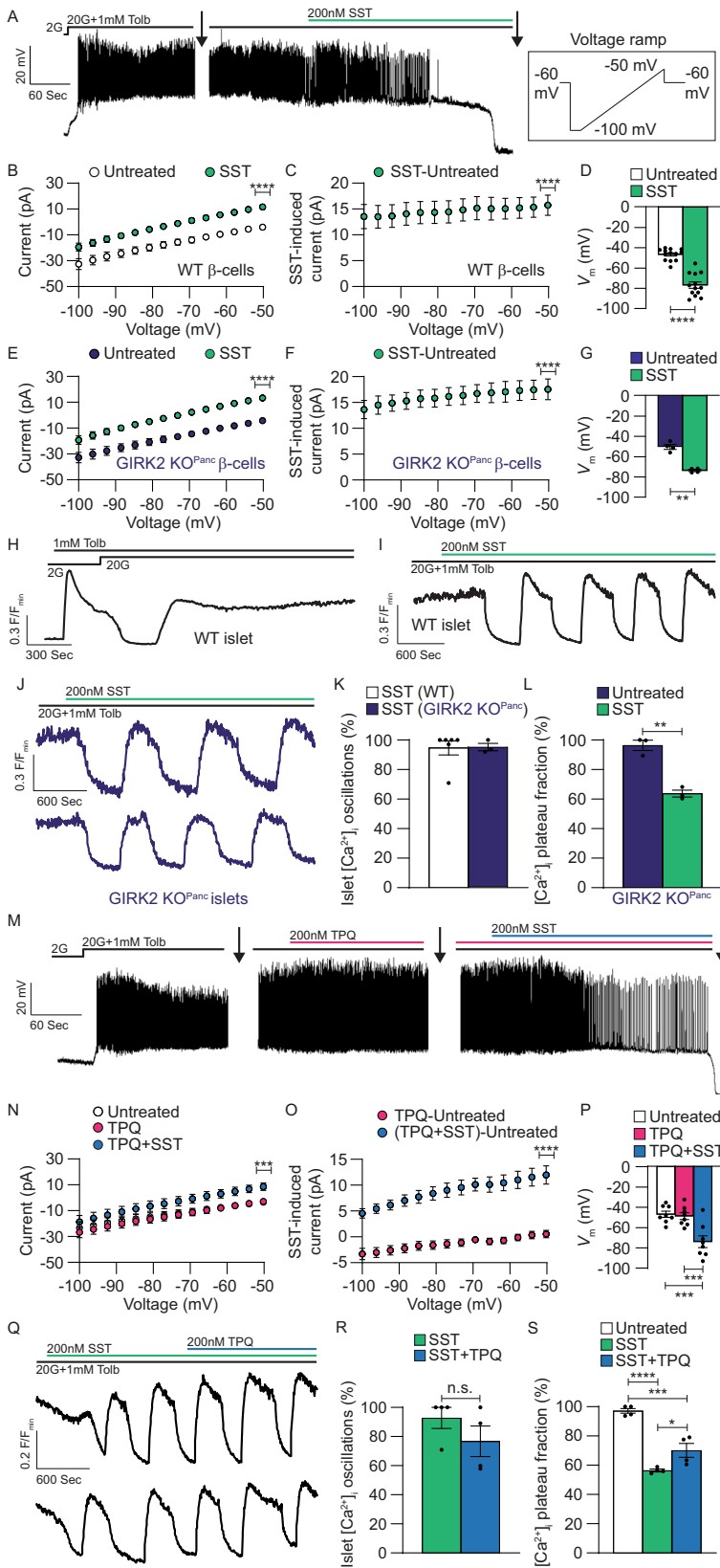

**Fig. 1 | SSTR-mediated β-cell currents are not due to GIRK channel activation.**
**A** Representative WT β-cell $V_m$ recording showing a typical SST response in the presence of 20 mM glucose (20G) + tolbutamide (Tolb). Whole-cell β-cell currents were measured (indicated by arrows) in response to a voltage ramp protocol (inset). **B** WT β-cell currents before (white) and after SST (green; $n = 11$). **C** SST-induced WT β-cell currents (green; $n = 11$). **D** WT β-cell $V_m$ before (white) and after SST (green; $n = 13$). **E** GIRK2 KO$^{Panc}$ β-cell currents before (dark blue) and after SST (green; $n = 4$). **F** SST-induced GIRK2 KO$^{Panc}$ β-cell currents (green; $n = 4$). **G** GIRK2 KO$^{Panc}$ β-cell $V_m$ before (dark blue) and after SST (green; $n = 4$). **H** Representative WT islet $[Ca^{2+}]_i$ response (F/F$_{min}$) to Tolb and 20G. **I** Representative WT islet SST $[Ca^{2+}]_i$ response. **J** Representative GIRK2 KO$^{Panc}$ islet SST $[Ca^{2+}]_i$ responses. **K** Percentage of WT islets (white; $n = 6$) and GIRK2 KO$^{Panc}$ islets (dark blue; $n = 3$) displaying SST-induced $[Ca^{2+}]_i$ oscillations. **L** GIRK2 KO$^{Panc}$ islet $[Ca^{2+}]_i$ plateau fraction before (dark blue) and after SST (green; $n = 3$). **M** Representative WT β-cell $V_m$ recording showing typical tertiapin-Q (TPQ) and SST responses. **N** WT β-cell currents before treatment (white), after TPQ (pink), and after TPQ + SST (blue; $n = 8$). **O** TPQ-induced (pink) and TPQ + SST-induced WT β-cell currents (green; $n = 8$). **P** WT β-cell $V_m$ before treatment (white), after TPQ (pink), and after TPQ + SST (blue; $n = 8$). **Q** Representative WT islet SST and TPQ $[Ca^{2+}]_i$ responses. **R** Percentage of WT islets displaying SST-induced $[Ca^{2+}]_i$ oscillations before (green) and after TPQ (blue; $n = 4$). **S** WT islet $[Ca^{2+}]_i$ plateau fraction before treatment (white), after SST (green), and after SST + TPQ (blue; $n = 4$). Statistical analysis was conducted using paired two-sided two-sample $t$ tests (**B, D, E, G, O**), unpaired two-sided two-sample $t$ tests (**K, L, R**), one-sample $t$ tests (**C, F**), or one-way ANOVA with Šidák's post-hoc multiple comparisons tests (**N, P, S**); *$P < 0.05$, **$P < 0.01$, ***$P < 0.001$, and ****$P < 0.0001$. Source data and exact $P$ values are provided as a Source Data file.

(DZ) during NKA inhibition (e.g., no extracellular K$^+$, Oua) decreased β-cell $[Ca^{2+}]_i$ by $82.5 \pm 15.7\%$ (Fig. 2B, E; $P < 0.01$), which demonstrates that NKA-mediated $V_m$ depolarization does not prevent $V_m$ hyperpolarization by K$^+$ channels. Following sustained increases in islet $[Ca^{2+}]_i$ that result from NKA inhibition (0 mM extracellular K$^+$), SST-induced $[Ca^{2+}]_i$ oscillations were rapidly restored by supplementation with 5 mM extracellular K$^+$ (Fig. 2F), thus it is unlikely that sustained $V_m$ depolarization resulting from NKA inhibition causes irreversible β-cell damage (Fig. 2F).

As NKA function is regulated by cAMP-dependent signaling pathways[38,39], we also measured mouse islet $[cAMP]_i$ along with $[Ca^{2+}]_i$ (Fig. 2A, B). Cross-correlation analysis of SST-mediated islet $[Ca^{2+}]_i$ and $[cAMP]_i$ oscillations revealed a negative correlation between the two (Fig. 2G; max correlation coefficient: $-0.46 \pm 0.02$) with $[Ca^{2+}]_i$ oscillations preceding $[cAMP]_i$ oscillations by approximately 100 ms. This suggests that $[cAMP]_i$ oscillations dynamically regulate β-cell NKA function. Furthermore, outward β-cell currents induced with 200 nM SST at −80 mV were completely inhibited by removal of extracellular K$^+$; this strongly suggests that SST-induced β-cell currents are mediated by NKAs (Fig. 2H, I). Lastly, we confirmed that SST does not induce β-cell currents or hyperpolarize $V_m$ in the absence of extracellular K$^+$ (Supplementary Fig. 2). Taken together, these data establish that SST-induced NKA activation decreases β-cell $[Ca^{2+}]_i$. Moreover, our results show that G$_{i/o}$-GPCR signaling induces oscillations in both $[Ca^{2+}]_i$ and $[cAMP]_i$, which likely results from oscillations in NKA activity.

NKA activity would be predicted to maintain low β-cell $[Na^+]_i$, therefore, we simultaneously measured islet $[Na^+]_i$ and $[Ca^{2+}]_i$. Cross-correlation analysis of SST-induced islet $[Na^+]_i$ and $[Ca^{2+}]_i$ oscillations demonstrated that changes in islet $[Na^+]_i$ closely follow $[Ca^{2+}]_i$ (Fig. 3A, B; in the presence of 20 mM glucose and 1 mM tolbutamide; max correlation coefficient: $0.67 \pm 0.08$). There was also a strong cross-correlation between islet $[Na^+]_i$ and $[Ca^{2+}]_i$ during glucose-stimulated (9 mM) $[Ca^{2+}]_i$ oscillations (Fig. 3C, D; max correlation coefficient: $0.75 \pm 0.11$). These findings show that islet $[Na^+]_i$ oscillates in response to SST, which may indicate that SSTR regulation of β-cell NKA function is also oscillatory. However, other Na$^+$ permeable ion channels could be involved as well. For example, β-cells express Na$^+$-permeable TRPM4 and TRPM5 channels that are activated by $[Ca^{2+}]_i$ and would be expected to facilitate Na$^+$ influx[51]. Although voltage-dependent Na$^+$ (Na$_V$) channels are also expressed in β-cells, these channels are largely inactive at voltages above −50 mV[52,53], and are thus unlikely to account for sustained (5–10 min) SST-mediated increases in islet $[Na^+]_i$.

It is established that elevations in $[cAMP]_i$ inhibit NKA activity[21] and we observed a correlation between increasing $[cAMP]_i$ and termination of SST-induced islet $[Ca^{2+}]_i$ oscillations (Fig. 2A, B). Thus, 5 µM forskolin (FSK) was employed to increase islet $[cAMP]_i$ and assess its impact on SST-induced β-cell NKA activity. FSK blocked β-cell NKA currents induced by 200 nM SST (Fig. 3E–G) and prevented $V_m$ hyperpolarization (Fig. 3H). Furthermore, SST-induced $[Ca^{2+}]_i$ oscillations persisted in only $15.3 \pm 6.2\%$ of islets following FSK treatment compared with SST alone ($84.8 \pm 6.2\%$ decrease; Fig. 3I, J; $P < 0.0001$); after FSK treatment islet $[Ca^{2+}]_i$ plateau fraction also increased by $40.5 \pm 2.9\%$ compared to with SST alone (Fig. 3I, K; $P < 0.0001$). These results suggest that increases in $[cAMP]_i$ can block SST-induced islet $[Ca^{2+}]_i$ oscillations in-part by inhibiting β-cell NKA activity.

## NKA activation is a conserved mechanism for G$_{i/o}$-GPCR control of β-cell $[Ca^{2+}]_i$

β-cells express a number of G$_{i/o}$-GPCRs in addition to SSTRs including α2A ADRs[11–15]. Thus, we examined whether β-cell NKA activation is a conserved mechanism for G$_{i/o}$ signaling-mediated control of islet $[Ca^{2+}]_i$. In the presence of 20 mM glucose and 1 mM tolbutamide islets did not oscillate until treated with 200 nM of the α-ADR activator clonidine (Clon)[54,55], after which all exhibited $[Ca^{2+}]_i$ oscillations (Fig. 4A, B; $P < 0.0001$) and a $64.0 \pm 3.3\%$ decrease in $[Ca^{2+}]_i$ plateau fraction (Fig. 4A, C; $P < 0.0001$). Treatment with 150 µM Oua terminated Clon-induced $[Ca^{2+}]_i$ oscillations in $94.9 \pm 3.0\%$ of islets (Fig. 4A, B; $P < 0.0001$) and increased $[Ca^{2+}]_i$ plateau fraction by $61.1 \pm 3.4\%$ (Fig. 4A, C; $P < 0.0001$), which was indistinguishable from islets before Clon treatment. These findings indicate that stimulation of NKA activity is a conserved mechanism for G$_{i/o}$-GPCR control of β-cell $[Ca^{2+}]_i$.

SSTRs and ADRs have also been shown to control α-cell function[16,56,57], which is predicted to impact β-cell Ca$^{2+}$ handling. Therefore, G$_{i/o}$-coupled Designer Receptors Exclusively Activated by Designer Drugs (DREADDs) driven by an optimized RIP were employed to selectively activate G$_{i/o}$ signaling in β-cells (βG$_{i/o}$-DREADDs; Fig. 4D). In the presence of 20 mM glucose and 1 mM tolbutamide islets expressing βG$_{i/o}$-DREADDs displayed no $[Ca^{2+}]_i$ oscillations. Following treatment with 10 µM clozapine N-oxide (CNO) $[Ca^{2+}]_i$ oscillations were observed in $88.5 \pm 9.0\%$ of islets (Fig. 4D, E) and islet $[Ca^{2+}]_i$ plateau fraction was decreased by $36.5 \pm 2.0\%$ (Fig. 4D, F; $P < 0.0001$). Interestingly, only a small subset of β-cells expressed G$_{i/o}$-DREADDs in each islet (based on mCherry fluorescence), suggesting that electrical coupling between β-cells amplifies G$_{i/o}$ signaling-induced islet $[Ca^{2+}]_i$ oscillations. After treatment with 150 µM Oua, $[Ca^{2+}]_i$ oscillations ceased in $93.4 \pm 4.3\%$ of islets (Fig. 4D, E; $P < 0.01$) and islet $[Ca^{2+}]_i$ plateau fraction increased $35.7 \pm 1.8\%$ (Fig. 4D, F, $P < 0.0001$), which was indistinguishable from islets before CNO treatment. These data confirm that under depolarizing conditions direct stimulation of β-cell G$_{i/o}$ signaling transiently hyperpolarizes $V_m$ by activating NKAs.

Paracrine activation of β-cell G$_s$-coupled glucagon-like peptide-1 receptors (GLP1Rs) and glucagon receptors (GCGRs) by glucagon secreted from α-cells helps to maintain β-cell $[cAMP]_i$ levels[58]. Because SST is a potent inhibitor of glucagon secretion, we next set out to determine if some SST-mediated changes in islet $[Ca^{2+}]_i$ oscillations were due to altered paracrine signaling from α-cells. Following treatment with 500 nM of the GLP1R antagonist exendin-3 (9–39) (Ex9) $[Ca^{2+}]_i$ oscillations were observed in only $18.7 \pm 18.7\%$ of islets and islet $[Ca^{2+}]_i$ plateau fraction was unaffected (Fig. 4G–I; in the presence of 20 mM glucose and 1 mM tolbutamide). Subsequent

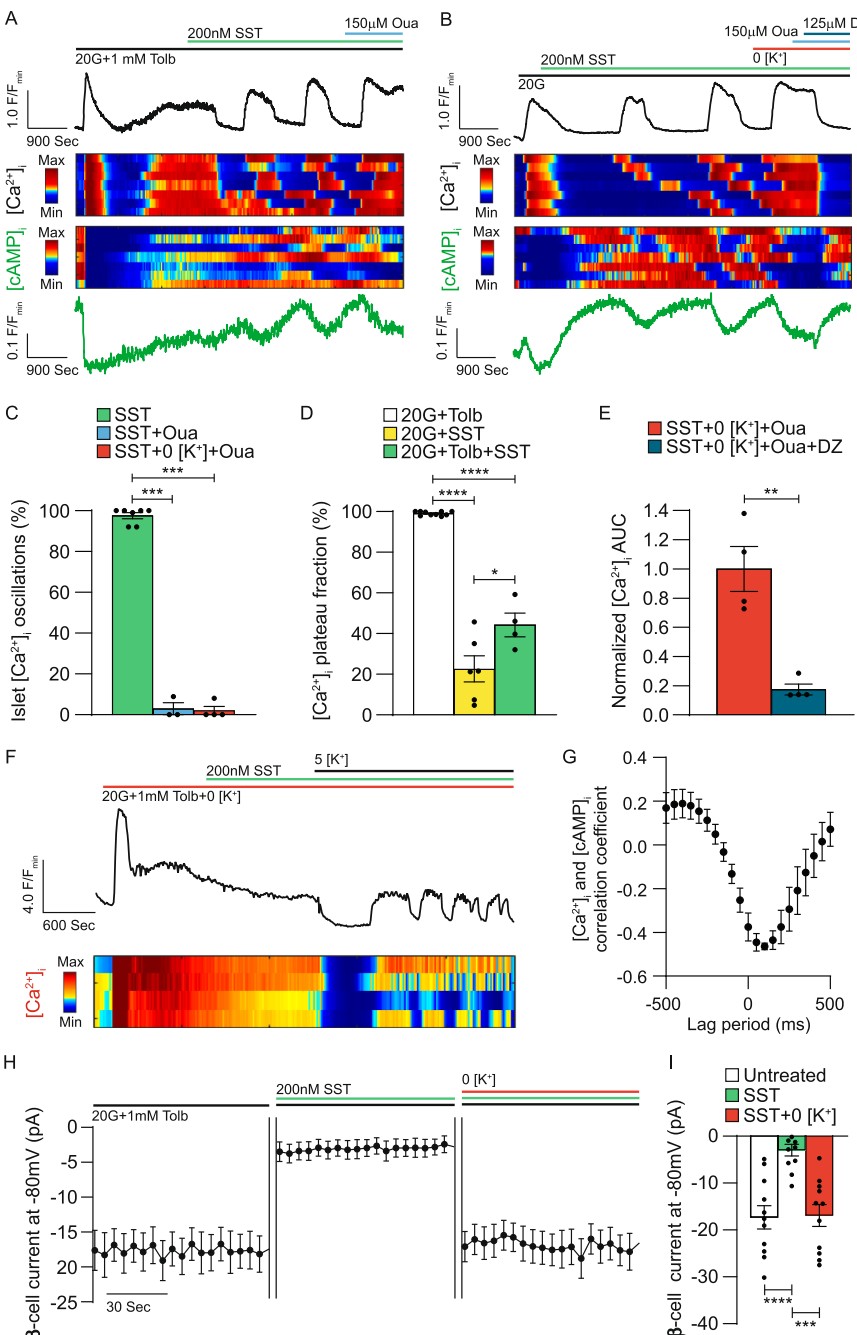

**Fig. 2 | SST-induced activation of β-cell NKAs generates islet [Ca²⁺]ᵢ and [cAMP]ᵢ oscillations. A** Representative WT islet jRGECO1a [Ca²⁺]ᵢ (top, black) and cAMPr [cAMP]ᵢ (bottom, green) responses (F/F$_{min}$) to SST and ouabain (Oua). Heatmaps illustrating typical islet [Ca²⁺]ᵢ (middle, upper) and [cAMP]ᵢ (middle, lower) responses. **B** Representative WT islet jRGECO1a [Ca²⁺]ᵢ and cAMPr [cAMP]ᵢ responses to SST, 0 mM extracellular K⁺ (0 [K⁺]), Oua, and diazoxide (DZ) in the presence of 20 mM glucose (20G). Heatmaps illustrating typical islet [Ca²⁺]ᵢ and [cAMP]ᵢ responses. **C** Percentage of WT islets displaying [Ca²⁺]ᵢ oscillations in response to SST (green; $n = 7$), SST + Oua (light blue; $n = 3$), and SST + 0 [K⁺] + Oua; orange; $n = 4$). **D** WT islet [Ca²⁺]ᵢ plateau fraction with 20G + Tolb (white; $n = 10$), 20 G + SST (yellow; $n = 6$), and 20G + Tolb + SST (green; $n = 4$). **E** WT islet [Ca²⁺]ᵢ AUC

(average of ≥10 min) normalized to 20G + SST + 0 [K⁺] + Oua before (orange) and after DZ (blue; $n = 4$). **F** Representative WT islet jRGECO1a [Ca²⁺]ᵢ (top) response to SST and 5 mM K⁺ (5 [K⁺]) in the presence of 20G + 0 [K⁺]. Heatmap illustrating typical islet [Ca²⁺]ᵢ responses (bottom). **G** Cross-correlation analysis of WT islet [Ca²⁺]ᵢ and [cAMP]ᵢ ($n = 4$). **H** WT β-cell currents recorded at −80 mV before treatment, after SST, and after removal of extracellular K⁺ ($n = 11$). **I** WT β-cell currents before treatment (white), after SST (green), and after removal of extracellular K⁺ (red; $n = 11$). Statistical analysis was conducted using an unpaired two-sided two-sample $t$ test (**E**) or one-way ANOVA with Šidák's post-hoc multiple comparisons tests (**C, D, I**); *$P < 0.05$, **$P < 0.01$, ***$P < 0.001$, and ****$P < 0.0001$. Source data and exact $P$ values are provided as a Source Data file.

treatment with 200 nM SST stimulated [Ca²⁺]ᵢ oscillations in all islets and decreased [Ca²⁺]ᵢ plateau fraction by 57.3 ± 8.5% (Fig. 4G–I; $P < 0.001$). Even under conditions of GLP1R blockade glucagon may still elevate β-cell [cAMP]ᵢ through GCGR signaling. Therefore, we assessed whether selective α-cell SSTR signaling could influence islet [Ca²⁺]ᵢ oscillations. For this, we utilized the selective SSTR2 agonist L-

054,264[16] to activate SSTR2s, which are exclusively expressed in mouse α-cells and are required for SST-mediated suppression of glucagon secretion[12,16,59]. Treatment with 500 nM L-054,264, did not stimulate islet [Ca²⁺]ᵢ oscillations and had no effect on islet [Ca²⁺]ᵢ plateau fraction (Fig. 4J–L; in the presence of 20 mM glucose and 1 mM tolbutamide). Taken together, these results suggest that G$_{i/o}$-

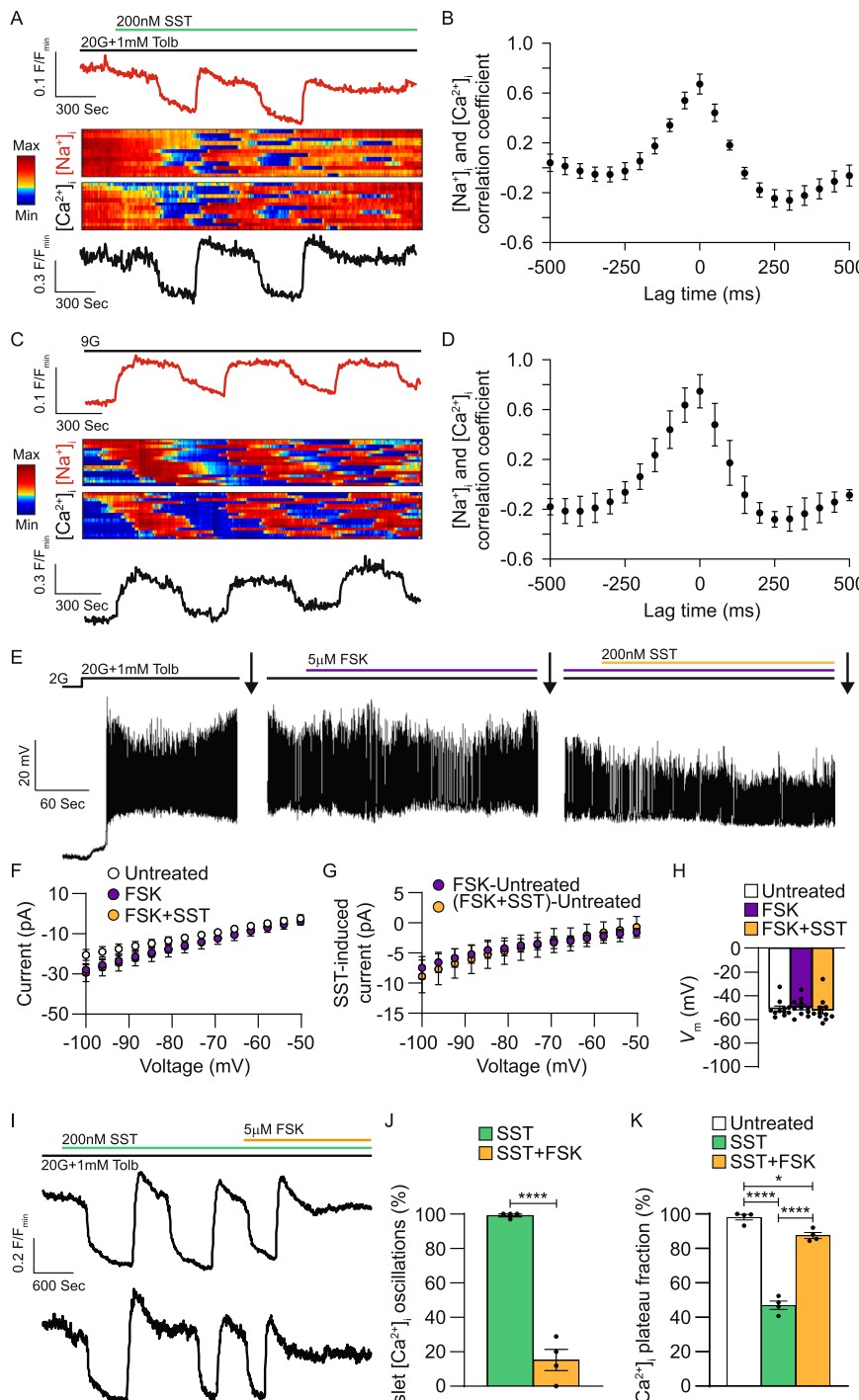

**Fig. 3 | NKA-mediated islet [Ca²⁺]ᵢ oscillations are inhibited by forskolin-induced increases in [cAMP]ᵢ. A** Representative WT islet ING-2 [Na⁺]ᵢ (top, red) and Fura Red AM [Ca²⁺]ᵢ (bottom, black) responses (F/F$_{min}$) to SST. Heatmaps illustrating typical islet [Na⁺]ᵢ (middle, upper) and [Ca²⁺]ᵢ (middle, lower) responses. **B** Cross-correlation analysis of WT islet [Na⁺]ᵢ and [Ca²⁺]ᵢ in the presence of 20 mM glucose (20G) + tolbutamide (Tolb) + SST ($n = 4$). **C** Representative WT islet [Na⁺]ᵢ and [Ca²⁺]ᵢ responses to 9 mM glucose (9G). Heatmaps illustrating typical islet [Na⁺]ᵢ and [Ca²⁺]ᵢ responses. **D** Cross-correlation analysis of WT islet [Na⁺]ᵢ and [Ca²⁺]ᵢ with 9G ($n = 3$). **E** Representative WT β-cell $V_m$ recording showing typical forskolin (FSK) and SST responses. Whole-cell β-cell currents were measured (indicated by arrows) in response to a voltage ramp protocol (see Fig. 1A inset). **F** WT β-cell currents

before treatment (white), after FSK (purple), and after FSK + SST (light orange; $n = 11$). **G** FSK-induced (purple) and FSK + SST-induced WT β-cell currents (light orange; $n = 11$). **H** WT β-cell $V_m$ before treatment (white), after FSK (purple), and after FSK + SST (light orange; $n = 11$). **I** Representative WT islet SST and FSK [Ca²⁺]ᵢ responses. **J** Percentage of WT islets displaying [Ca²⁺]ᵢ oscillations in response to SST (green) and SST + FSK (light orange; $n = 4$). (K) WT islet [Ca²⁺]ᵢ plateau fraction before treatment (white), after SST (green), and after SST + FSK (light orange; $n = 4$). Statistical analysis was conducted using paired two-sided two-sample $t$ tests (**F, G**), unpaired two-sided two-sample $t$ tests (**J**), or one-way ANOVA with Šidák's post-hoc multiple comparisons tests (**H, K**); *$P < 0.05$ and ****$P < 0.0001$. Source data and exact $P$ values are provided as a Source Data file.

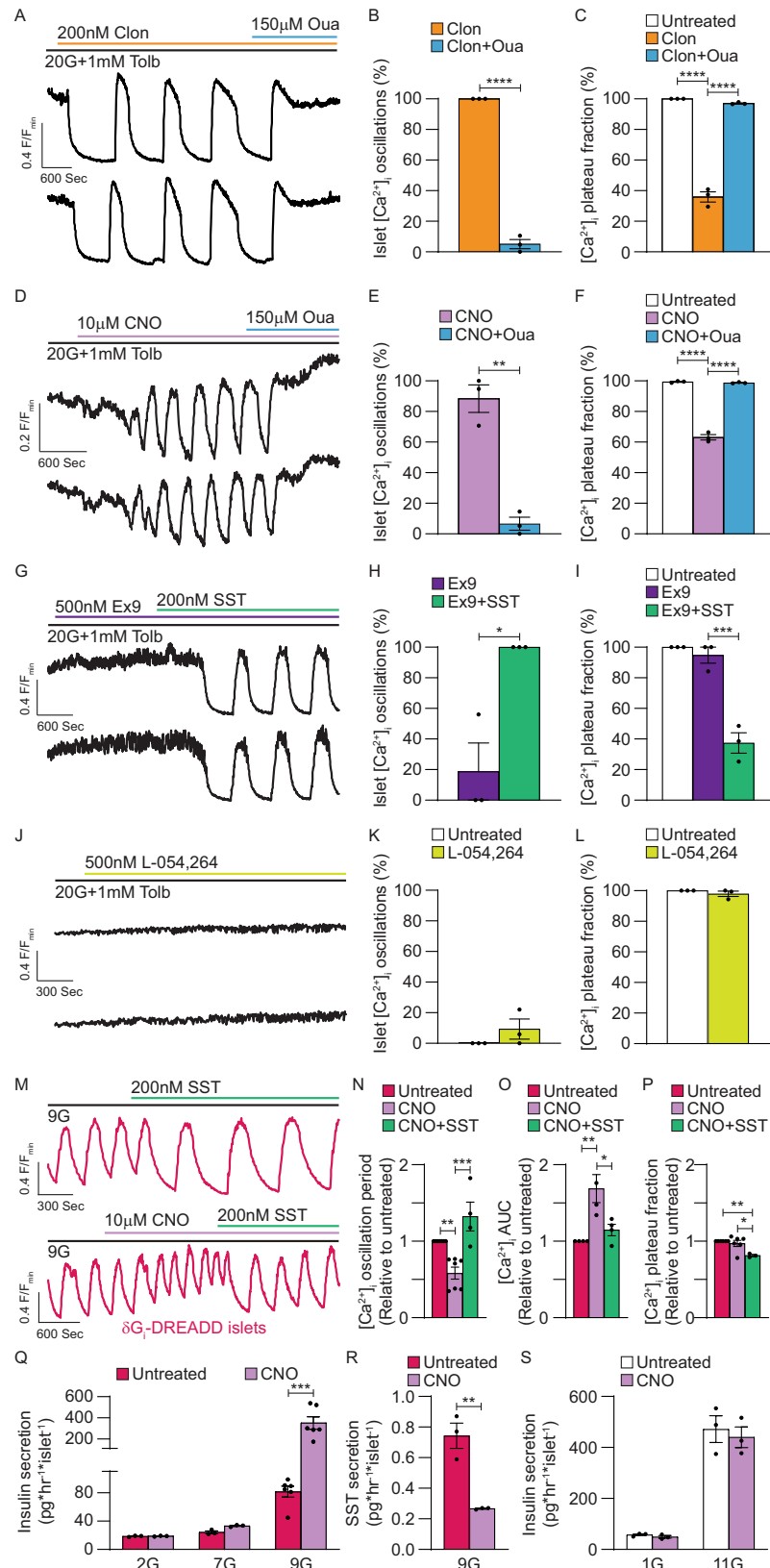

coupled ligands primarily initiate islet $[Ca^{2+}]_i$ oscillations by directly activating β-cell $G_{i/o}$-GPCRs (i.e., SSTRs, α-ADRs) rather than by diminishing paracrine activation of β-cell $G_s$-GPCRs (i.e., GLP1Rs, GCGRs).

We next sought to determine if endogenous islet SST controls $[Ca^{2+}]_i$ oscillations; this was accomplished utilizing islets from transgenic mice selectively expressing $G_{i/o}$-coupled DREADDs in δ-cells (δ$G_i$-DREADDs) to inhibit SST secretion during glucose-stimulated $[Ca^{2+}]_i$ oscillations (Fig. 4M). Following a 20 min equilibration period with 9 mM glucose, δ$G_i$-DREADD islets exhibited stable $[Ca^{2+}]_i$ oscillations that did not change significantly over a period of 40 min (Supplementary Fig. 3A, B). Activation of δ-cell $G_{i/o}$ signaling with 10 μM

**Fig. 4 | Activation of β-cell NKAs is a conserved mechanism for $G_{i/o}$-GPCR control of islet $Ca^{2+}$ handling. A** Representative WT islet clonidine (Clon) and ouabain (Oua) $[Ca^{2+}]_i$ responses (F/$F_{min}$). **B** Percentage of WT islets displaying $[Ca^{2+}]_i$ oscillations with Clon (orange) and Clon + Oua (blue; $n = 3$). **C** WT islet $[Ca^{2+}]_i$ plateau fraction before treatment (white), after Clon (orange), and after Clon + Oua (blue; $n = 3$). **D** Representative β$G_{i/o}$-DREADD-expressing WT islet CNO and Oua $[Ca^{2+}]_i$ responses. **E** Percentage of β$G_{i/o}$-DREADD-expressing islets displaying $[Ca^{2+}]_i$ oscillations in response to CNO (pink) and CNO + Oua (blue; $n = 3$). **F** β$G_{i/o}$-DREADD-expressing WT islet $[Ca^{2+}]_i$ plateau fraction before treatment (white), after CNO (pink), and after CNO + Oua (blue; $n = 3$). **G** Representative WT islet Exendin-3 (9–39) (Ex9) and SST $[Ca^{2+}]_i$ responses. **H** Percentage of WT islets displaying $[Ca^{2+}]_i$ oscillations in response to Ex9 (purple) and Ex9 + SST (green; $n = 3$). (**I**) WT islet $[Ca^{2+}]_i$ plateau fraction before treatment (white), after Ex9 (purple), and after Ex9 + SST (green; $n = 3$). **J** Representative WT islet L-054,264 $[Ca^{2+}]_i$ responses. **K** Percentage of WT islets displaying $[Ca^{2+}]_i$ oscillations before (white) and after L-

054,264 (yellow; $n = 3$). (**L**) WT islet $[Ca^{2+}]_i$ plateau fraction before (white) and after L-054,264 (yellow; $n = 3$). **M** Representative δ$G_{i/o}$-DREADD islet CNO/SST $[Ca^{2+}]_i$ responses. **N** Normalized δ$G_{i/o}$-DREADD islet $[Ca^{2+}]_i$ oscillation period before treatment (magenta; $n = 7$), after CNO (pink; $n = 7$), and after CNO + SST (green; $n = 4$). **O** Normalized δ$G_{i/o}$-DREADD islet $[Ca^{2+}]_i$ AUC before treatment (magenta), after CNO (pink), and after CNO + SST (green; $n = 4$). **P** Normalized δ$G_{i/o}$-DREADD islet $[Ca^{2+}]_i$ plateau fraction before treatment (magenta; $n = 6$), after CNO (pink; $n = 6$), and after CNO + SST (green; $n = 3$). **Q** δ$G_{i/o}$-DREADD islet insulin secretion without CNO (magenta) and with CNO (pink; 2G: $n = 3$, 7 G: $n = 3$, 9 G: $n = 6$). **R** SST secretion from δ$G_{i/o}$-DREADD islets without CNO (magenta) and with CNO (pink; 9G: $n = 3$). **S** WT islet insulin secretion without CNO (white) and with CNO (pink; 1G: $n = 3$, 11 G: $n = 3$). Statistical analysis was conducted using unpaired two-sided two-sample $t$ tests (**B**, **E**, **H**, **K**, **L**, **R**), or one-way ANOVA with Šidák's post-hoc multiple comparisons tests (**C**, **F**, **I**, **N–Q**, **S**); *$P < 0.05$, **$P < 0.01$, ***$P < 0.001$, and ****$P < 0.0001$. Source data and exact $P$ values are provided as a Source Data file.

CNO in the presence of 9 mM glucose decreased the period of islet $[Ca^{2+}]_i$ oscillations by 41.7 ± 7.7% (Fig. 4N; $P < 0.01$) and increased islet $[Ca^{2+}]_i$ area under the curve (AUC) by 68.8 ± 18.3% (Fig. 4O; $P < 0.01$) relative to the period prior to addition of CNO. However, δ$G_i$-DREADD activation had no effect on islet $[Ca^{2+}]_i$ plateau fraction (Fig. 4P). The effects of δ-cell $G_{i/o}$ signaling were reversed following treatment with 200 nM SST (Fig. 4M); islet $[Ca^{2+}]_i$ oscillation period increased by 74.1 ± 20.4% (Fig. 4N; $P < 0.001$), $[Ca^{2+}]_i$ AUC decreased by 54.0 ± 19.8% (Fig. 4O; $P < 0.05$), and $[Ca^{2+}]_i$ plateau fraction decreased by 15.8 ± 4.4% (Fig. 4P; $P < 0.05$). Similarly, 200 nM SST decreased δ$G_i$-DREADD islet $[Ca^{2+}]_i$ AUC by 21.2 ± 11.4% in the absence of CNO (Supplementary Fig. 3C, D; $P < 0.0001$).

As stimulation of δ-cell $G_{i/o}$ signaling enhanced β-cell $Ca^{2+}$ influx and accelerated $[Ca^{2+}]_i$ oscillation frequency, we examined the effect of δ-cell $G_{i/o}$-GPCR activation on SST and insulin secretion. δ$G_i$-DREADD activation had no effect on insulin secretion at 2 or 7 mM glucose; however, at 9 mM glucose δ-cell $G_{i/o}$ signaling increased insulin secretion from 81.5 ± 7.7 to 352.2 ± 59.4 pg insulin $h^{-1} \cdot islet^{-1}$ (Fig. 4Q; $P < 0.0001$). Importantly, δ$G_i$-DREADD activation decreased SST secretion from 0.743 ± 0.082 to 0.267 ± 0.003 pg SST $h^{-1} islet^{-1}$ (Fig. 4R; $P < 0.01$) under these conditions. Insulin secretion from WT islets was not affected by CNO at either 1 or 11 mM glucose (Fig. 4S). These findings show that δ-cell SST secretion regulates islet $Ca^{2+}$ handling and insulin secretion under physiological conditions. The data also suggest that SSTR-mediated activation of β-cell NKAs slows glucose-stimulated $[Ca^{2+}]_i$ oscillations and resulting pulsatile insulin secretion.

## $G_{i/o}$-GPCRs control human islet $Ca^{2+}$ handling by increasing β-cell NKA activity

As SSTR signaling hyperpolarizes human β-cell $V_m$ and inhibits voltage-dependent $Ca^{2+}$ ($Ca_V$) channel activity[16], we went on to examine if NKAs contribute to this effect. Transcriptional analysis indicates that human β-cells express high levels of *ATP1A1* transcript (gene encoding the NKA α1 subunit)[11–13]. Immunofluorescence staining of human pancreatic sections confirmed that insulin positive β-cells also stain positive for NKA α1, and revealed a predominantly cell membrane-restricted localization (Fig. 5A). Interestingly, other islet cells stained positive for NKA α1, which may indicate that NKA serves additional roles in human pancreatic α- and/or δ-cells.

To assess whether NKAs influence SSTR control of human β-cell $Ca^{2+}$ handling a genetically encoded $[Ca^{2+}]_i$ indicator expressed selectively in human β-cells (RIP-GCaMP6s) was utilized[60]. Treatment with 400 nM SST at 7 mM glucose decreased β-cell $[Ca^{2+}]_i$ AUC by 37.2 ± 0.8% (Fig. 5B, C; $P < 0.05$) and $[Ca^{2+}]_i$ plateau fraction by 36.4 ± 2.9% (Fig. 5B, D; $P < 0.01$). Moreover, 400 nM SST prompted significant decreases in human islet $[Ca^{2+}]_i$ AUC, which was reduced by 79.9 ± 16.4% in the absence of extracellular $K^+$ (Fig. 5E, F; in the presence of 20 mM glucose and 1 mM tolbutamide; $P < 0.05$). As human islets express a number of GIRK channel transcripts[16], TPQ was utilized

to assess the contribution of human β-cell GIRK channels to $G_{i/o}$-GPCR-mediated $V_m$ hyperpolarization. When GIRK channels were inhibited with 200 nM TPQ, SST mediated human islet $[Ca^{2+}]_i$ decreases that were indistinguishable from human islets not treated with TPQ (Fig. 5G, H). Importantly, SST hyperpolarized β-cell $V_m$ from −56.8 ± 2.8 to −69.7 ± 4.1 mV (Fig. 5I, J; $P < 0.01$) and activated outward currents that were inhibited by removal of extracellular $K^+$ (current amplitude at −100 mV: 2.0 ± 0.2pA; Fig. 5K–M; $P < 0.0001$). SST-induced human β-cell currents were outward at voltages below the equilibrium potential of $K^+$ (Fig. 5M), which again suggests $Na^+$ movement through NKAs. Taken together, these data indicate that SSTR signaling increases human β-cell NKA activity resulting in $V_m$ hyperpolarization and decreased islet $[Ca^{2+}]_i$.

## β-cell NKA activity is regulated by PKA and tyrosine kinase signaling

FSK-mediated elevations in $[cAMP]_i$ influence β-cell function primarily by stimulating PKA signaling[58,61], which has been shown to decrease NKA activity[38,39,45]; thus, PKA was pharmacologically blocked with 10 μM H89 to investigate its role in $G_{i/o}$-GPCR control of β-cell electrical activity and $Ca^{2+}$ handling. Treatment with H89 after establishment of SST-induced islet $[Ca^{2+}]_i$ oscillations hyperpolarized β-cell $V_m$, which prevented $Ca^{2+}$ influx through $Ca_V$ channels (Fig. 6A); as a result, islet $[Ca^{2+}]_i$ AUC was decreased by 84.7 ± 3.6% (Fig. 6A, B; $P < 0.001$) and $[Ca^{2+}]_i$ plateau fraction was reduced by 99.5 ± 0.3% (Fig. 6A, C; $P < 0.0001$) compared to SST alone. Furthermore, H89 abolished FSK-mediated increases in islet $[Ca^{2+}]_i$ AUC (Fig. 6A, B) as well as $[Ca^{2+}]_i$ plateau fraction (Fig. 6A, C). These findings suggest that PKA serves as a negative regulator of β-cell NKA function; however, H89 has off-target effects[62] and PKA inhibition likely affects the function of other β-cell ion channels[63]. Thus, to limit potential confounding effects, β-cell $V_m$ was clamped in a hyperpolarized state with 125 μM DZ and islet $[Na^+]_i$ measured as an indicator of $G_{i/o}$-GPCR-mediated β-cell NKA activity (Fig. 6D–F). Under these conditions, SST increased islet $Na^+$ efflux by 43.9 ± 5.8% (Fig. 6D, G; $P < 0.01$) compared to before treatment, which was inhibited by Oua (Fig. 6D, G; $P < 0.001$). SST-induced islet $Na^+$ efflux (39.1 ± 7.4% increase; Fig. 6E, H; $P < 0.05$) was also inhibited following GLP1R activation with 400 nM liraglutide (Lira; Fig. 6E, H; $P < 0.01$), indicating that $G_s$ protein-coupled receptor ($G_s$-GPCR)-mediated increases in β-cell $[cAMP]_i$ block NKA activity. Interestingly, in islets pretreated with 1 μM of myristoylated protein kinase inhibitory peptide 14–22 amide (PKI) for 1 h to inhibit PKA activity, SST increased $Na^+$ efflux by 63.9 ± 2.6% (Fig. 6F, I; $P < 0.01$) compared to before treatment, but Lira-mediated inhibition of SST-induced islet $Na^+$ efflux was completely blocked (Fig. 6F, I; $P < 0.01$). Treatment with 150 μM Oua was able to prevent SST-induced islet $Na^+$ efflux even in the presence of PKI (Fig. 6F, I; $P < 0.01$). These findings indicate that β-cell NKAs are likely inhibited by cAMP-dependent PKA activity.

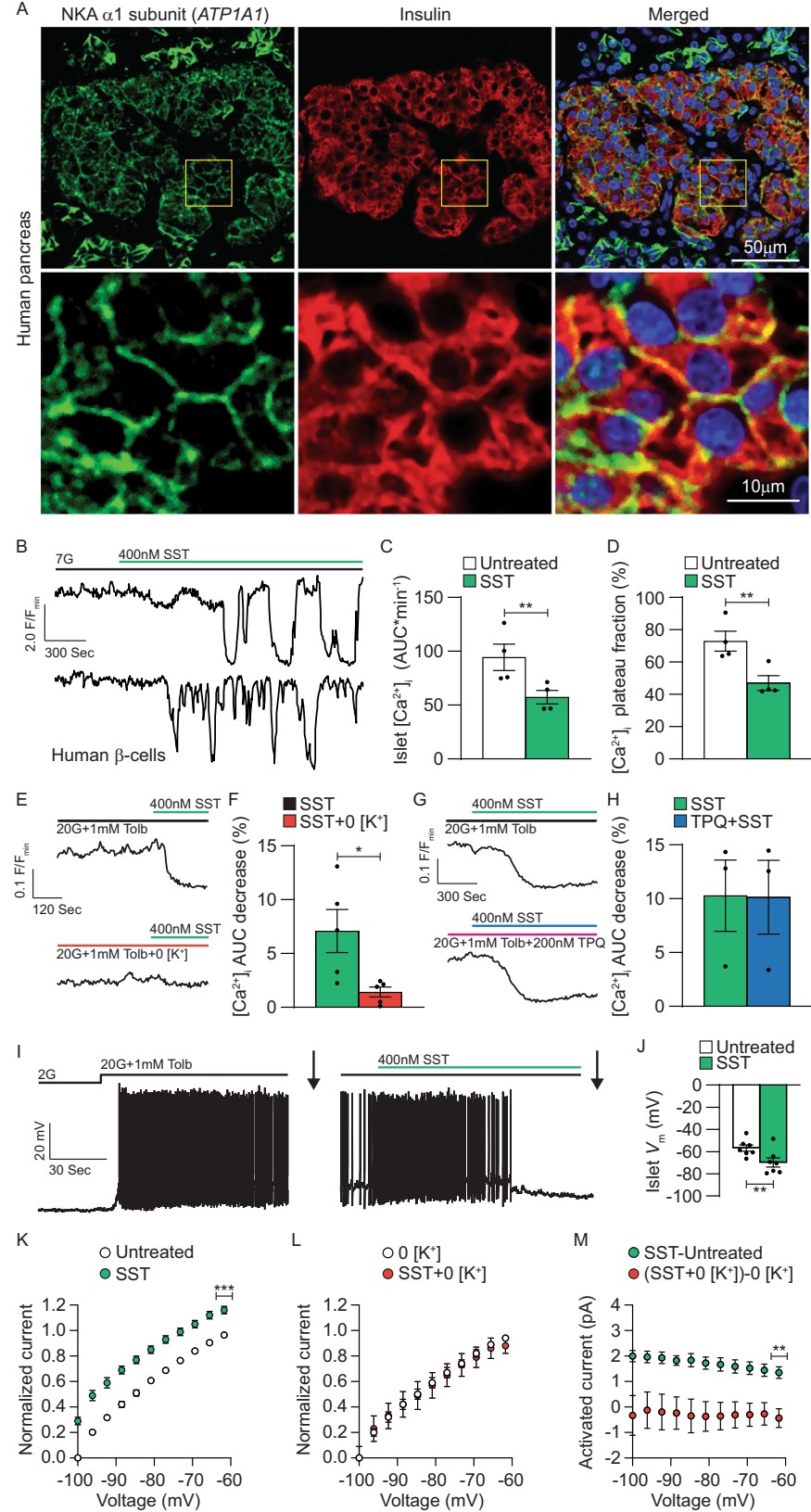

It has been shown that SSTR signaling stimulates STK activity[42], which interacts with and enhances NKA function[43,44,46]. Therefore, a STK inhibitor dasatinib (Dasa) was employed to investigate the role of STK signaling in $G_{i/o}$-GPCR control of β-cell NKA function (Fig. 7A). All islets initially displayed SST-induced $[Ca^{2+}]_i$ oscillations, which

decreased to $37.6 \pm 9.8\%$ following treatment with 100 nM Dasa (Fig. 7A, B; in the presence of 20 mM glucose and 1 mM tolbutamide; $P < 0.01$). Dasa also increased islet $[Ca^{2+}]_i$ plateau fraction by $40.7 \pm 3.1\%$ compared to SST alone (Fig. 7A, C; $P < 0.001$). The average size of islets that oscillated in the presence of Dasa was $71.0 \pm 8.6\%$ larger than islets

**Fig. 5 | G$_{I/o}$-GPCRs regulate human β-cell electrical excitability by stimulating NKA activity. A** Top row: Representative immunofluorescent staining of a healthy human pancreatic section for NKA α1 subunits (green), insulin (red), and a merged image of the two showing colocalization (yellow; staining representative of pancreas sections from 4 human donors). Bottom row: Magnification of the corresponding areas outlined with yellow boxes above. **B** Representative human β-cell SST [Ca$^{2+}$]$_i$ responses (F/F$_{min}$; within intact human islets) at 7 mM glucose (7G). **C** Human β-cell [Ca$^{2+}$]$_i$ AUC (average of ≥15 min) at 7G before (white) and after SST (green; $n = 4$). **D** Human β-cell [Ca$^{2+}$]$_i$ plateau fraction at 7G before (white) and after SST (green; $n = 4$). **E** Representative human islet SST [Ca$^{2+}$]$_i$ responses with (top) and without extracellular K$^+$ (0 [K$^+$]; bottom). **F** SST-induced decrease in human islet [Ca$^{2+}$]$_i$ AUC (sum of 5 min; relative to before treatment) with (green) and without extracellular K$^+$ (red; $n = 5$). **G** Representative human islet SST [Ca$^{2+}$]$_i$ responses in the absence (top) and the presence of tertiapin-Q (TPQ; bottom). **H** SST-induced decrease in human islet [Ca$^{2+}$]$_i$ AUC (sum of 5 min; relative to before treatment) in the absence (green) and the presence of TPQ (blue; $n = 3$). **I** Representative human β-cell $V_m$ recording showing a typical SST response. Whole-cell β-cell currents were measured (indicated by arrows) in response to a voltage ramp protocol (see Fig. 1A inset). **J** Human β-cell $V_m$ before (white) and after SST (green; $n = 7$). **K** Normalized human β-cell currents ($I/I_{min}$; $I_{min}$ = minimum current before SST) before (white) and after SST (green; $n = 8$). **L** Normalized human β-cell currents with 0 [K$^+$] before (white) and after SST (red; $n = 5$). **M** SST-induced human β-cell currents with (green; $n = 8$) and without extracellular K$^+$ (red; $n = 5$). Statistical analysis was conducted using paired two-sided two-sample $t$ tests (**C**, **D**, **F**, **H**, and **J**–**L**) or an unpaired two-sided two-sample $t$ test (**M**); *$P < 0.05$, **$P < 0.01$, and ***$P < 0.001$. Source data and exact $P$ values are provided as a Source Data file.

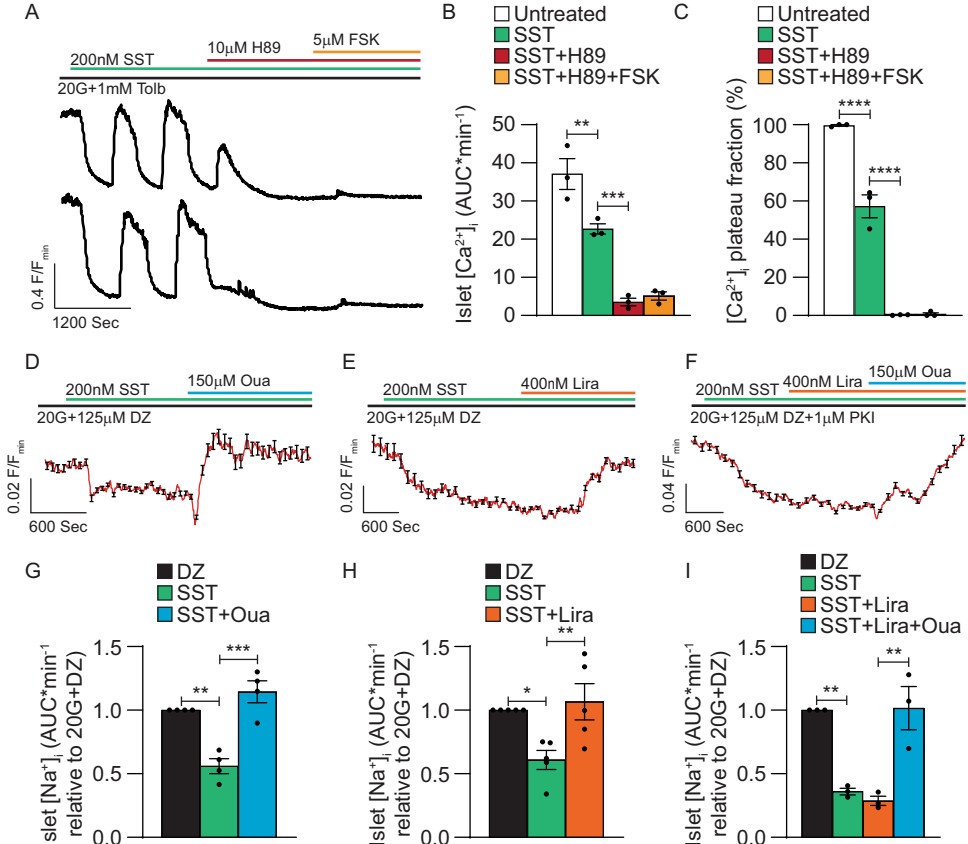

**Fig. 6 | G$_{I/o}$-GPCR-mediated β-cell NKA activity is inhibited by PKA.**
**A** Representative WT islet SST, H89, and forskolin (FSK) [Ca$^{2+}$]$_i$ responses (F/F$_{min}$). **B** WT islet [Ca$^{2+}$]$_i$ AUC (average of ≥5 min) before treatment (white), after SST (green), after SST + H89 (red), and after SST + H89 + FSK (orange; $n = 3$). **C** WT islet [Ca$^{2+}$]$_i$ plateau fraction before treatment (white), after SST (green), after SST + H89 (red), and after SST + H89 + FSK (orange; $n = 3$). **D** Normalized WT islet SST and ouabain (Oua) [Na$^+$]$_i$ responses (F/F$_{min}$; $n = 19$) in the presence of 20 mM glucose (20G) + diazoxide (DZ). **E** Normalized WT islet SST and liraglutide (Lira) [Na$^+$]$_i$ responses ($n = 28$) in the presence of 20G + DZ. **F** Normalized WT islet SST, Lira, and Oua [Na$^+$]$_i$ responses ($n = 22$) in the presence of 20G + DZ + PKI. **G** WT islet [Na$^+$]$_i$ AUC (average of ≥5 min; normalized to 20G + DZ) before treatment (black), after SST (green), and after SST + Oua (blue; $n = 4$). **H** WT islet [Na$^+$]$_i$ AUC (average of ≥5 min; normalized to 20G + DZ) before treatment (black), after SST (green), and after SST + Lira (orange; $n = 5$). **I** WT islet [Na$^+$]$_i$ AUC (average of ≥5 min; normalized to 20G + DZ + PKI) before treatment (black), after SST (green), after SST + Lira (orange), and after SST + Lira + Oua (blue; $n = 3$). Statistical analysis was conducted using one-way ANOVA with Šidák's post-hoc multiple comparisons tests (**B**, **C**, **G**–**I**); *$P < 0.05$, **$P < 0.01$, ***$P < 0.001$, and ****$P < 0.0001$. Source data and exact $P$ values are provided as a Source Data file.

that stopped oscillating (Fig. 7D; $P < 0.01$), which may indicate that bifurcated islet Ca$^{2+}$ responses were due in-part to incomplete penetration of Dasa into larger islets.

As SSTR signaling activates Src homology region 2 domain-containing phosphatase-2 (Shp2) that in turn stimulates STK function[42,64], a Shp2 inhibitor NSC 87877 (NSC) was utilized to assess whether this pathway regulates β-cell NKAs (Fig. 7E). All islets exhibited [Ca$^{2+}$]$_i$ oscillations in response to 200 nM SST, which decreased to $53.5 \pm 6.1\%$ following treatment with 5 μM NSC (Fig. 7E, F; in the presence of 20 mM glucose and 1 mM tolbutamide; $P < 0.01$). Furthermore, NSC increased islet [Ca$^{2+}$]$_i$ plateau fraction by $39.9 \pm 6.8\%$ compared to SST alone (Fig. 7E, G; $P < 0.01$). As with Dasa, the average size of islets that continued oscillating with NSC was $75.7 \pm 16.7\%$ larger than islets that stopped oscillating (Fig. 7H; $P < 0.05$), which again suggests only partial Shp2 inhibition in larger islets.

Insulin receptor tyrosine kinases can stimulate STK activity as well as augment NKA function[35,65]. Therefore, we examined if insulin enhances β-cell NKA activity. Islet [Ca$^{2+}$]$_i$ imaging experiments were

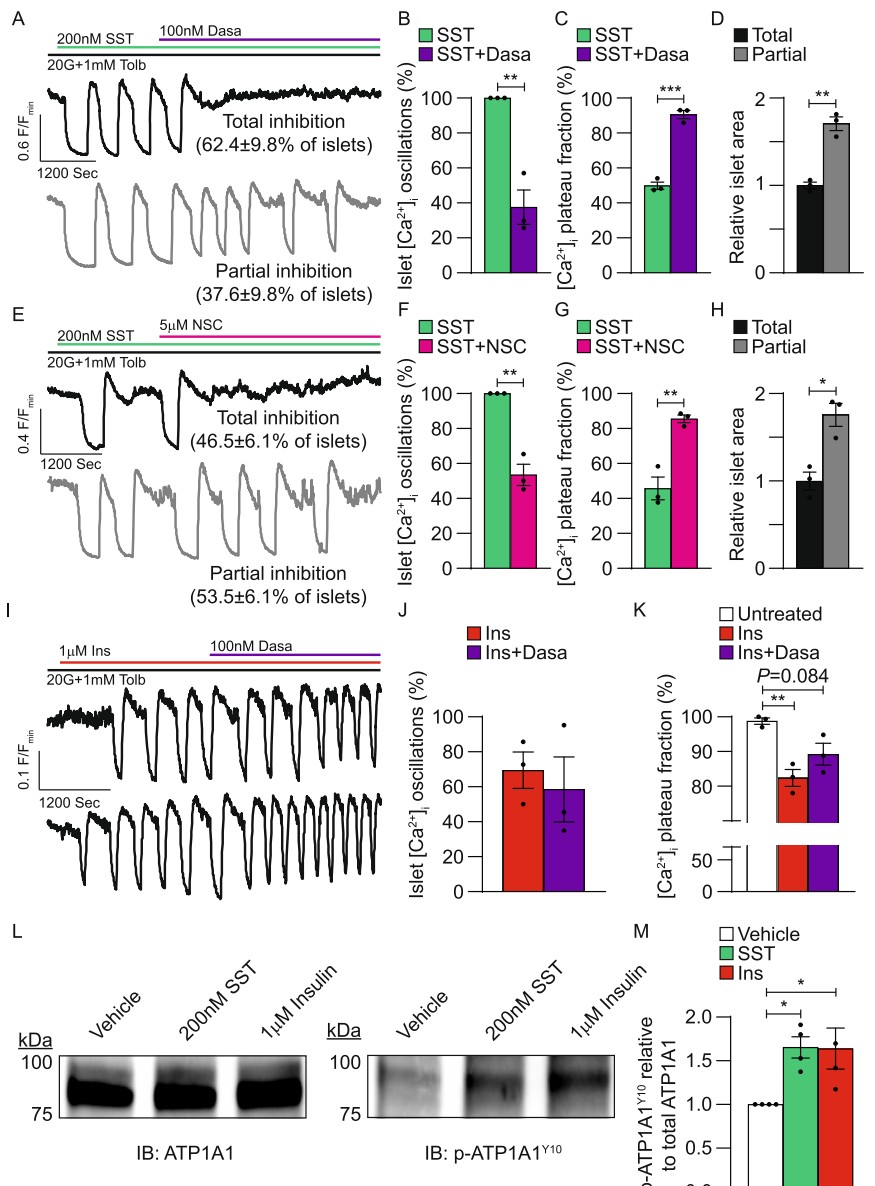

**Fig. 7 | $G_{i/o}$-GPCR-mediated β-cell NKA activity depends on tyrosine kinase signaling. A** Representative WT islet SST and dasatinib (Dasa) $[Ca^{2+}]_i$ responses (F/ $F_{min}$). Insets indicate the percentage of islets exhibiting each type of Dasa $[Ca^{2+}]_i$ response. **B** Percentage of WT islets displaying $[Ca^{2+}]_i$ oscillations in response to SST (green) and SST + Dasa (purple; $n = 3$). **C** WT islet $[Ca^{2+}]_i$ plateau fraction after SST (green) and after SST + Dasa (purple; $n = 3$). **D** Relative size of WT islets displaying total (black) or partial (gray) Dasa-mediated inhibition of SST-induced $[Ca^{2+}]_i$ oscillations ($n = 3$). **E** Representative WT islet SST and NSC 87877 (NSC) $[Ca^{2+}]_i$ responses. Insets indicate the percentage of islets exhibiting each type of NSC $[Ca^{2+}]_i$ response. **F** Percentage of WT islets displaying $[Ca^{2+}]_i$ oscillations in response to SST (green) and SST + NSC (pink; $n = 3$). **G** WT islet $[Ca^{2+}]_i$ plateau fraction after SST (green) and after SST + NSC (pink; $n = 3$). **H** Relative size of WT islets displaying total (black) or partial (gray) NSC-mediated inhibition of SST-induced $[Ca^{2+}]_i$ oscillations ($n = 3$). **I** Representative WT islet insulin (Ins) and Dasa

$[Ca^{2+}]_i$ responses. **J** Percentage of WT islets displaying $[Ca^{2+}]_i$ oscillations in response to Ins (red) and Ins + Dasa (purple; $n = 3$). **K** WT islet $[Ca^{2+}]_i$ plateau fraction before treatment (white), after Ins (red), and after Ins + Dasa (purple; $n = 3$). **L** Representative immunoblots (IBs) of WT islet cell lysates that were treated for 15 min with vehicle ($H_2O$), SST, or Ins at 25 °C. IBs were probed for total NKA (ATP1A1; left) and NKA phosphorylated at Y10 (p-ATP1A1$^{Y10}$; right). p-ATP1A1$^{Y10}$ bands were normalized to corresponding ATP1A1 bands; normalized p-ATP1A1$^{Y10}$ bands from SST-/Ins-treated islets were then normalized to p-ATP1A1$^{Y10}$ bands from vehicle-treated islets. **M** Average vehicle-normalized islet p-ATP1A1$^{Y10}$ relative to total ATP1A1 ($n = 4$). Statistical analysis was conducted using unpaired two-sided two-sample $t$ tests (**B–D**, **F–H**, **J**) or one-way ANOVA with Šidák's post-hoc multiple comparisons tests (**K**, **M**); *$P < 0.05$, **$P < 0.01$, and ***$P < 0.001$. Source data and exact $P$ values are provided as a Source Data file.

carried out at 25 °C to suppress endogenous insulin secretion[66]. Treatment with 1 μM insulin-induced $[Ca^{2+}]_i$ oscillations in 69.5 ± 10.4% of islets (Fig. 7I, J; in the presence of 20 mM glucose and 1 mM tolbutamide) and decreased islet $[Ca^{2+}]_i$ plateau fraction by 16.4 ± 2.6% (Fig. 7K; $P < 0.01$). Subsequent addition of 100 nM Dasa had no effect on the percentage of islets displaying insulin-induced $[Ca^{2+}]_i$ oscillations (58.6 ± 18.6%; Fig. 7I, J). Islet $[Ca^{2+}]_i$ plateau fraction trended higher after Dasa treatment compared to insulin alone (6.9 ± 4.0%

increase; Fig. 7K); however, the change was not significant. These results suggest that β-cell NKA function is augmented by autocrine insulin receptor signaling, but that this effect is not mediated by STKs. Furthermore, these findings again indicate that once stimulated β-cell NKA function is oscillatory in nature, independent of the mechanism of activation.

Phosphorylation of the NKA α1 subunit by tyrosine kinases has been shown to regulate pump activity[45,67–69]. Thus, to elucidate the

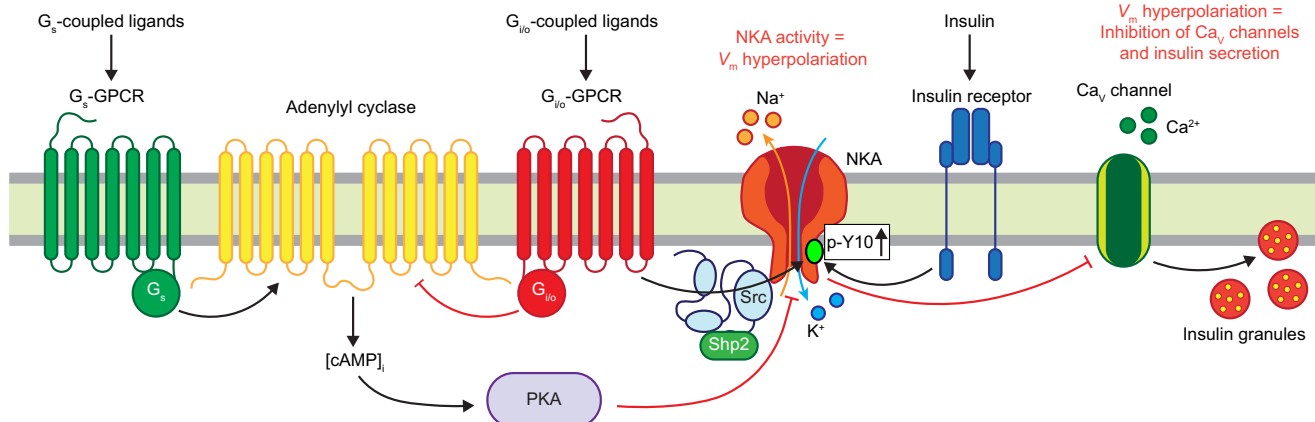

**Fig. 8 | Model illustrating the mechanisms that regulate β-cell NKA function.** Overview of stimulatory and inhibitory receptor-mediated signaling pathways that tune β-cell NKA activity. $G_{i/o}$-GPCR signaling cyclically hyperpolarizes β-cell $V_m$ via STK-mediated phosphorylation of NKAs as well as by decreasing $[cAMP]_i$ and PKA activity. Other tyrosine kinases (e.g., insulin receptors) also phosphorylate and activate β-cell NKAs. Furthermore, stimulation of $G_s$-GPCRs diminishes β-cell NKA function by increasing $[cAMP]_i$ and PKA activity.

mechanism underlying tyrosine kinase control of β-cell NKA function, mouse islets were maximally stimulated with 20 mM glucose and 1 mM tolbutamide at 25 °C then treated with 200 nM SST, 1 μM insulin, or a vehicle control (H₂O); islet cell lysates were immunoblotted for total NKA α1 protein as well as for NKA α1 phosphorylated at a putative tyrosine kinase phosphorylation site (tyrosine 10 (Y10))[45,68,69]. Phosphorylation of NKA α1 at residue Y10 (p-ATP1A1[Y10]) increased by 65.3 ± 12.3% in islets treated with SST ($P < 0.05$) and by 64.0 ± 23.4% in islets treated with insulin (Fig. 7L, M; $P < 0.05$) relative to vehicle controls. As previous studies have shown that phosphorylation of NKA α1 at residue Y10 increases NKA activity[45,68,69], this result suggests that $G_{i/o}$-GPCR (and insulin receptor) signaling augments β-cell NKA function in-part through phosphorylation of the NKA α1 subunit.

## Discussion

Islet $G_{i/o}$-GPCR signaling modulates β-cell $[Ca^{2+}]_i$ oscillations, which in turn regulate pulsatile insulin secretion[6,18,21]. Here, we elucidate the mechanistic underpinnings of β-cell $G_{i/o}$ signaling-induced $V_m$ hyperpolarization. We found that $G_{i/o}$-GPCR activation stimulates β-cell NKAs, which hyperpolarizes $V_m$ independently of $K_{ATP}$ channel activity and leads to decreased islet $[Ca^{2+}]_i$ oscillation frequency. Furthermore, STK activation was required for $G_{i/o}$ signaling-induced activation of β-cell NKAs, whereas NKA-mediated $V_m$ hyperpolarization was inhibited by cAMP-dependent PKA activation. This indicates that multiple signaling modalities converge to control NKA function and thus islet $[Ca^{2+}]_i$ handling. These results demonstrate the importance of δ-cell SST secretion in controlling islet $[Ca^{2+}]_i$ oscillations through SSTR-mediated regulation of β-cell NKA activity. Moreover, this work reveals the conserved (SSTRs, α2A ADRs) mechanism for $G_{i/o}$-GPCR control of β-cell NKA function. Therefore, the data establish that regulation of β-cell NKA activity by $G_{i/o}$ signaling plays a critical role in tuning the frequency of islet $[Ca^{2+}]_i$ oscillations, and thus the kinetics of insulin secretion.

Numerous studies have shown that $G_{i/o}$ signaling hyperpolarizes β-cell $V_m$[16,18,19,36,54,55], which has been attributed to $G_{i/o}$-GPCR-mediated activation of a $K^+$ conductance. Stimulation of GIRK channel activity accounts for $G_{i/o}$ signaling-induced $V_m$ hyperpolarization in many tissues[37] and this has also been proposed for β-cells[16,36]. While a small SST-induced GIRK-like conductance has been recorded in human β-cells[16], elegant studies by Sieg et al. demonstrated that GIRK channels are not activated in mouse β-cells by ADR stimulation[19]. We also observed minimal activation of a SSTR-mediated GIRK conductance in mouse or human β-cells. However, as GIRK channel inhibition

increased islet $[Ca^{2+}]_i$ plateau fraction in islets undergoing SST-induced $[Ca^{2+}]_i$ oscillations, GIRK channels are predicted to play a conserved role in $G_{i/o}$-GPCR-mediated β-cell $V_m$ hyperpolarization. To further enhance β-cell GIRK channels responsible for this change in islet $[Ca^{2+}]_i$, $K^+$ was removed from the extracellular solution. Surprisingly, SST did not stimulate β-cell currents or hyperpolarize $V_m$ in the absence of extracellular $K^+$, suggesting that SST-induced $V_m$ hyperpolarization may not be due to a $K^+$ conductance. Furthermore, SST-activated currents were present in β-cells lacking GIRK2 channels, were not inhibited by TPQ, and interestingly were outward at voltages below the equilibrium potential of $K^+$, further establishing that a non-$K^+$ conductance is activated by β-cell SSTR signaling. As $G_{i/o}$-GPCR signaling activates NKAs in other tissues and removal of extracellular $K^+$ inhibits NKA activity, we considered NKAs a strong candidate for SST-induced β-cell $V_m$ hyperpolarization[38,40]. This was confirmed using the NKA inhibitor Oua, which blocked SST-induced islet $[Ca^{2+}]_i$ decreases. Moreover, epinephrine has been shown to hyperpolarize β-cell $V_m$ and stimulate NKAs in certain tissues[9,38], and indeed, our data confirmed that α2A-ADR signaling decreases β-cell $[Ca^{2+}]_i$ by activating NKAs. Taken together, these results suggest that while activation of both NKAs and GIRK channels control islet $[Ca^{2+}]_i$ plateau fraction, NKA activation serves a predominant and conserved role in $G_{i/o}$ signaling-mediated β-cell $V_m$ hyperpolarization.

The mechanisms underlying $G_{i/o}$-GPCR control of NKA activity are highly tissue-specific, but are strongly linked to the phosphorylation status of the NKA α subunit[38,39,45]. Our data show that β-cell NKA activity is inhibited by FSK- and GLP1R-mediated increases in $[cAMP]_i$ and that this effect depends on PKA activity. These data support previous findings indicating that $G_{i/o}$ signaling-induced islet $[Ca^{2+}]_i$ oscillations can be terminated by stimulating adenylyl cyclase (AC) activity, inhibiting phosphodiesterase (PDE) activity, or directly raising $[cAMP]_i$[21]. Furthermore, as cellular metabolism increases β-cell $[cAMP]_i$ levels[61,70,71], PKA-mediated NKA-inhibition would be predicted to be glucose-sensitive. These results strongly suggest that phosphorylation by PKA inhibits β-cell NKA activity.

β-cell $[cAMP]_i$ decreased only modestly immediately after SST treatment, which was likely a consequence of islet $[Ca^{2+}]_i$ being clamped at an elevated level. Under these conditions, β-cell $[cAMP]_i$ was greatly reduced even prior to SST treatment, presumably due in-part to cAMP degradation by $Ca^{2+}$-activated PDEs (i.e., PDE1) as well as by diminished cAMP generation by $Ca^{2+}$-inhibited ACs (i.e., AC6, AC9)[12,13,72,73]. Thus, it is probable that because $[cAMP]_i$ was already so low that $G_{i/o}$-GPCR activation could not reduce it further. Several minutes after SST treatment we observed β-cell $[cAMP]_i$ oscillations

## Table 1 | Human islet donors

| RRID | Age | BMI | Assays performed |
|---|---|---|---|
| SAMN20064638 | 66 | 29.9 | Electrophysiology |
| SAMN20209569 | 45 | 21.9 | Electrophysiology |
| SAMN20923891 | 45 | 50.6 | Ca²⁺ imaging |
| SAMN20926064 | 62 | 26.8 | Ca²⁺ imaging |
| SAMN21032331 | 40 | 27.4 | Ca²⁺ imaging and electrophysiology |
| SAMN21244110 | 53 | 31.2 | Electrophysiology |
| SAMN21855451 | 55 | 35.1 | Ca²⁺ imaging & electrophysiology |
| SAMN22021186 | 39 | 29 | Ca²⁺ imaging & electrophysiology |
| SAMN22818629 | 45 | 21.7 | Ca²⁺ imaging |
| SAMN22814513 | 26 | 29.2 | Ca²⁺ imaging |
| SAMN28450743 | 47 | 25.5 | Ca²⁺ imaging |
| SAMN28867622 | 36 | 29.6 | Ca²⁺ imaging |
| SAMN29005116 | 47 | 44.8 | Ca²⁺ imaging |

that were out-of-phase with $[Ca^{2+}]_i$ oscillations, which is consistent with $Ca^{2+}$-dependent inhibition of $[cAMP]_i$ accumulation (i.e., PDE1 activation, AC6/AC9 inhibition). Taken together, these findings suggest that $G_{i/o}$-GPCR signaling (a) activates β-cell NKAs leading to inhibition of $Ca^{2+}$ entry; (b) islet $[Ca^{2+}]_i$ decreases drive cAMP generation leading to PKA-mediated NKA inhibition, which results in $Ca^{2+}$ influx; (c) this in turn, reduces $[cAMP]_i$ and alleviates PKA inhibition of NKA, thus restoring tyrosine kinase-supported β-cell NKA $V_m$ hyperpolarization. While SST-induced β-cell $[Ca^{2+}]_i$ oscillations are regulated by $[cAMP]_i$ levels, there are other factors that could contribute to the oscillatory nature of $G_{i/o}$-GPCR-mediated β-cell NKA activity. For example, NKAs require sufficient $[Na^+]_i$ levels to function[74], thus NKA-mediated depletion of β-cell $[Na^+]_i$ during periods of sustained activity may lead to NKA inactivation allowing for $V_m$ depolarization and $[Ca^{2+}]_i$ influx. Furthermore, as SSTRs and other $G_{i/o}$-GPCRs undergo desensitization and internalization when continually exposed to ligands[75], receptor recycling would be predicted to play a role in β-cell NKA oscillations. However, when islet $[Ca^{2+}]_i$ was clamped at a low level, SST-induced prolonged non-oscillatory decreases in islet $[Na^+]_i$ that were inhibited with conditions that increase islet cAMP, suggesting that β-cell NKAs are primarily inhibited by cAMP-dependent PKA activity. Although, these findings indicate that $G_{i/o}$-GPCR-mediated β-cell NKA activity is tuned by fluctuations in PKA signaling, additional kinases that modulate NKA function (i.e., PKC, EPAC, PKG) could also be impacted by changes in $[cAMP]_i$[76–80]. Thus, future studies are required to more completely determine the mechanisms that control the oscillatory nature of $G_{i/o}$-GPCR-mediated β-cell NKA activity.

Although islet $[cAMP]_i$ oscillations emerged over time, induction of SST-induced islet $[Ca^{2+}]_i$ oscillations despite minimal initial $[cAMP]_i$ decreases suggested that cAMP was not the only signal controlling $G_{i/o}$-GPCR-mediated β-cell NKA oscillations. While cAMP-dependent changes in PKA activity modulate β-cell NKA function, $G_{i/o}$-GPCRs also stimulate cAMP-independent protein kinases that have been shown to influence NKA activity. For example, STK and Shp2, which activates STK, interact with SSTRs and are activated by SSTR signaling[42,64]. Moreover, STKs interact with and augment NKA activity[43,46]. As STK and Shp2 inhibition greatly diminished NKA-mediated decreases of islet $[Ca^{2+}]_i$, SST-induced STK signaling is predicted to facilitate activation of β-cell NKAs. Interestingly, the STK inhibitor used in these studies, Dasa, which is FDA approved for the treatment of Philadelphia chromosome-positive chronic myeloid leukemia, has been shown to decrease blood glucose levels in numerous clinical studies, most prominently in diabetic patients[81–83]. This suggests the possibility that Dasa increases human islet insulin secretion by inhibiting NKAs, and thus enhancing glucose-mediated $Ca^{2+}$ entry.

Insulin receptors that interact with and control STK signaling are also tyrosine kinases and have been shown to stimulate NKAs in a variety of tissues[35,65,84]. Our findings confirmed that insulin enhances β-cell NKA function; however, insulin-induced islet $[Ca^{2+}]_i$ oscillations were only modestly affected by Dasa treatment, indicating that insulin stimulates β-cell NKAs independently of STK signaling, possibly through direct phosphorylation of NKA α1 subunits. Phosphorylation of NKA α1 by tyrosine kinases has been shown to augment NKA function[45,67,85] and indeed, phosphorylation of islet NKA α1 residue Y10 increased following SST and insulin treatment. This establishes that $G_{i/o}$-GPCR-mediated β-cell NKA activation is due in-part to phosphorylation by tyrosine kinases. Therefore, it will be important to further examine the mechanisms underlying tyrosine kinase regulation of β-cell NKA function and its role in tuning islet $[Ca^{2+}]_i$ oscillations as well as GSIS.

SST secretion becomes defective during the pathogenesis of diabetes[33,86]; thus, control of β-cell NKA function by SSTR signaling would be expected to be perturbed during T2D. Other islet cell types, including α- and δ-cells, also express high levels of NKA α1 subunit transcript[11–13], which is supported by our immunofluorescence staining of human pancreatic sections that showed NKA α1 subunit expression in insulin-negative islet cells (Fig. 5A). Furthermore, numerous NKA β and γ subunit transcripts as well as several $G_{i/o}$-GPCRs are expressed in α-cells (i.e., SST, D2-like DRDs, and α1-ADRs) and δ-cells (i.e., D2-like DRDs, and α2-ADRs)[11–13,16,30,87]. Moreover, NKAs have been shown to regulate α-cell $V_m$ in response to fatty acid metabolism[88]. Thus, NKA control of plasma membrane $V_m$ would be expected to influence $Ca^{2+}$ handling in these other islet cell types and may help explain why GIRK channel inhibition fails to completely inhibit SST-induced α-cell $V_m$ hyperpolarization[57]. Lastly, expression of specific NKA subunits, such as FXYD2, are altered in T2D human islets[11,89] and in leptin receptor deficient diabetic mouse islets[90]. Thus, understanding the role(s) that NKA serves in all islet cell types under physiological and diabetic conditions will illuminate critical features of islet function and disfunction.

In summary, we identified a conserved $G_{i/o}$-coupled mechanism for controlling β-cell $Ca^{2+}$ entry, and thus insulin secretion in response to numerous $G_{i/o}$-GPCR ligands that have been shown to limit insulin secretion (SST, epinephrine). Moreover, we demonstrated that endogenous SSTR signaling tunes islet $[Ca^{2+}]_i$ oscillations and presumably pulsatile insulin secretion by activating β-cell NKAs. Finally, we determined that $G_{i/o}$ signaling stimulates β-cell NKA function by activating tyrosine kinases (STKs, insulin receptors). Taken together, these findings reveal an essential and conserved NKA-mediated mechanism governing $G_{i/o}$-coupled signals known to regulate insulin secretion that remained elusive for over half a century (Fig. 8).

## Methods
### Animals
All mice were 10- to 16-week old, age-matched males on a C57Bl6/J background (Stock #: 000664; The Jackson Laboratory (JAX), Bar Harbor, ME). Transgenic mice expressing $G_{i/o}$- DREADDs specifically in δ-cells (δ$G_i$ DREADDs) were generated by crossing mice expressing a mutant $G_{i/o}$-GPCR-P2A-mCitrine (B6.129-Gt(ROSA)$^{26Sortm1(CAG-CHRM4*,-mCitrine)Ute/J}$; Stock #: 026219; JAX) construct preceded by a loxP-flanked STOP cassette with mice expressing a *SST*-IRES-Cre (Stock #: 013044; JAX)[91,92]. All δ$G_i$ DREADD islets were prepared from mice heterozygous for *SST*-IRES-Cre as this transgene decreases endogenous SST expression in an allele dosage-dependent manner. Transgenic mice with pancreatic-specific knockout (KO) of *Kcnj6* (gene encoding GIRK2) were generated by crossing animals with a loxP-flanked *Kcnj6* exon 4 with mice expressing a *Pdx1*-Cre (GIRK2 KO$^{Panc}$; Stock #: 014647; JAX)[93,94]. All animals were housed in a Vanderbilt University IACUC (protocol # M1600063-01) approved facility on a 12-h light/dark cycle with

access to standard chow (Lab Diets, 5LOD) *ad libitum*. Mice were humanely euthanized by cervical dislocation followed by exsanguination. To preserve islet ion channel function mice were not treated with anesthesia.

## Human donors

All studies detailed here were approved by the Vanderbilt University Health Sciences Committee Institutional Review Board (IRB# 110164). Healthy human islets were provided from multiple isolation centers by the Integrated Islet Distribution Program (IIDP). Deidentified human donor information is provided in Table 1. The IIDP obtained informed consent for deceased donors in accordance with NIH guidelines prior to reception of human islets for our studies. The deidentified healthy human pancreas samples stained in Fig. 5 were obtained from the NCI funded Cooperative Human Tissue Network (CHTN) (https:// www. chtn.org/). Written consent was obtained for deceased donors by the CHTN prior to reception of human pancreatic tissue.

## Chemicals and reagents

Unless otherwise noted all chemicals and reagents were purchased from Sigma-Aldrich (St. Louis, MO) or Thermo Fisher (Waltham, MA). Clozapine N-oxide (CNO) was purchased from Hello Bio (Princeton, NJ). Clonidine hydrochloride (Clon), exendin-3 (9–39) amide (Ex9), forskolin (FSK), L-054,264, ouabain (Oua), and myristoylated PKI 14–22 amide (PKI) were purchased from R&D Systems (Minneapolis, MN). Tertiapin-Q (TPQ) was purchased from Alomone Labs (Jerusalem, Israel). Dasatinib (Dasa), H-89, and NSC 87877 were purchased from Cayman Chemical (Ann Arbor, MI). Liraglutide (Lira) was purchased from Novo Nordisk (Plainsboro, NJ). An optimized rat insulin promoter (RIP)[60] and the coding sequence of hM4D(Gi)-mCherry ($G_{i/o}$-DREADD-mCherry fusion protein; Plasmid #75033; Addgene, Watertown, MA)[95] were cloned into a pLenti6 lentiviral transfer plasmid and utilized to produce 3rd-generation lentiviruses (LVs) that express $G_{i/o}$-DREADDs selectively in β-cells (β$G_{i/o}$-DREADDs) as previously described[96].

## Islet isolation

Mouse pancreata were digested with collagenase P (Roche; Basel, Switzerland) and islets were isolated using density gradient centrifugation[53,97,98]; mouse islets were cultured in RPMI-1640 (Corning) media with 5.6 mM glucose supplemented with 15% FBS, 100 IU ml$^{-1}$ penicillin, and 100 mg ml$^{-1}$ streptomycin (RPMI) at 37 °C, 5% $CO_2$. Upon arrival, human islets were allowed to recover for at least 2 h in CMRL-1066 (Corning, Cleveland, TN) media containing 5.6 mM glucose and supplemented with 20% fetal bovine serum (FBS), 100 IU ml$^{-1}$ penicillin, 100 mg ml$^{-1}$ streptomycin, 2 mM Gluta-MAX, 2 mM HEPES, and 1 mM sodium pyruvate (CMRL) at 37 °C, 5% $CO_2$. Mouse and human islets were cultured in poly-D-lysine-coated 35 mm glass-bottomed dishes (CellVis, Mountain View, CA) and all experiments were conducted within 48 h.

## Patch-clamp electrophysiology

Patch electrodes (3–4 MΩ) were backfilled with intracellular solution containing (mM) 90.0 KCl, 50.0 NaCl, 1.0 MgCl$_2$, 10.0 EGTA, 10.0 HEPES, and 0.005 amphotericin B (adjusted to pH 7.2 with KOH). Mouse and human islets were patched in Krebs-Ringer HEPES buffer (2 mL; KRHB) containing (mM) 119.0 NaCl, 4.7 KCl, 2.0 CaCl$_2$, 1.2 MgSO$_4$, 1.2 KH$_2$PO$_4$, and 10.0 HEPES (pH 7.35 adjusted by NaOH) supplemented with indicated glucose concentrations; a perforated whole-cell patch-clamp technique was utilized to record β-cell membrane potential ($V_m$) in current-clamp mode using an Axopatch 200B amplifier with pCLAMP10 software (Molecular Devices)[99,100]. Islet cells that did not display electrical activity at 2 mM glucose were identified as β-cells. After a perforated patch configuration was established (seal resistance >1.0 GΩ; leak< 20.0 pA) $V_m$ depolarization and action potential (AP) firing were induced by exchanging the bath solution

with KRHB supplemented with 20 mM glucose and 1 mM tolbutamide. After AP firing was observed, the amplifier was switched to voltage-clamp mode; $V_m$ was held at −60 mV and the membrane voltage was ramped from −100 mV to −50 mV every 15 s for at least 3 min and the resulting β-cell currents recorded. The amplifier was then returned to current-clamp mode and $V_m$ recorded. Mouse and human islets were perifused with further treatments and changes in β-cell $V_m$ and currents measured as indicated in figure legends.

## Intracellular $Ca^{2+}$ and cAMP imaging

For simultaneous islet $[Ca^{2+}]_i$ and $[cAMP]_i$ imaging, mouse islets were transduced for 4 h with an adenovirus (AV) expressing a CMV-jRGECO1a-P2A-cAMPr construct (VectorBuilder, Chicago, IL) and cultured 24 h at 37 °C, 5% $CO_2$ prior to imaging. Alternatively, mouse islets were loaded with a Fura-2 AM $Ca^{2+}$ indicator (2 μM) for 30 min before the start of an experiment. Human islets were either transduced for 4 h with an AV expressing RIP-GCaMP6s (VectorBuilder) and cultured 48 h before imaging or loaded with a Cal-590 $Ca^{2+}$ indicator (10 μM; AAT Bioquest, Sunnyvale, CA) for 1 h prior to the start of a study. Before each experiment, islet culture media was replaced with 2 mL KRHB supplemented with indicated glucose concentrations; after 10 min, the islets were treated as detailed in figure legends. Mouse islet jRGECO1a (excitation (Ex): 561 nm; emission (Em): 620 ± 50 nm) and cAMPr fluorescence (Ex: 488 nm; Em: 525 ± 32 nm) were simultaneously measured every 5 s utilizing an LSM 780 multi-photon confocal microscope equipped with Zeiss Zen software (×20 magnification; LSM 780) as indicators of $[Ca^{2+}]_i$ and $[cAMP]_i$ respectively. Mouse islet Fura-2 AM fluorescence (Ex: 340 nm and 380 nm; Em: 510 ± 40 nm) was measured every 5 s with a Nikon Ti2 epifluorescence microscope equipped with a Prime 95B camera with 25 mm CMOS sensors and Nikon Elements software (×10 magnification; Nikon Ti2); the ratio of Fura-2 AM fluorescence excited at 340 nm and 380 nm was utilized as an indicator of $[Ca^{2+}]_i$. Human islet Cal-590 (Ex: 560 ± 20 nm; Em: 630 ± 37.5 nm) or β-cell GCaMP6s fluorescence (Ex: 488 nm; Em: 531 ± 48 nm) was measured every 5 s as an indicator of $[Ca^{2+}]_i$ utilizing the LSM 780 microscope (×20 magnification).

## Intracellular $Ca^{2+}$ and $Na^+$ imaging

For simultaneous islet $[Ca^{2+}]_i$ and $[Na]_i$ imaging, mouse islets were loaded with a Fura Red AM $Ca^{2+}$ indicator (5 μM; catalog #: F3021; Thermo Fisher) and an ION Natrium Green-2 (ING-2) $Na^+$ indicator (5 μM; catalog #: 2011F; Ion Biosciences, San Marcos, TX) for 1 h before the start of an experiment. Some experiments as indicated in figure legends were carried out with only ING-2. Before each experiment, islet culture media was replaced with KRHB supplemented with indicated glucose concentrations; after 10 min, the islets were treated as detailed in figure legends. Mouse islet Fura Red (Ex: 430 ± 12 nm and 500 ± 10 nm; Em: 700 ± 37.5 nm) fluorescence was measured every 5 s with the Nikon Ti2 (×10 magnification); the ratio of Fura Red fluorescence excited at 500 ± 10 nm and 430 ± 12 nm was utilized as an indicator of $[Ca^{2+}]_i$. Mouse islet ING-2 fluorescence (Ex: 500 ± 10 nm; Em: 535 ± 15 nm) was measured every 5 s as an indicator of $[Na]_i$.

## Immunofluorescence imaging

Paraffin-embedded human pancreas sections were processed and probed as previously described[96]. Following rehydration, sections were subject to Tris-EDTA-SDS antigen retrieval at 37 °C for 40 min[101]; pancreas sections were stained with primary antibodies (1:100 mouse anti-ATP1A1 (catalog #: MA3-928; Thermo Fisher) and 1:1000 guinea pig anti-insulin (catalog #: 20-IP35; Fitzgerald, North Acton, MA) followed by secondary antibodies (1:300 donkey anti-mouse Alexa Fluor 647 (catalog #: 715-606-150; Jackson ImmunoResearch, West Grove, PA) and 1:300 donkey anti-guinea pig Alexa Fluor 488 (catalog #: 706-

546-148; Jackson ImmunoResearch)). Immunofluorescence images were collected with a Zeiss LSM 710 META inverted confocal microscope (×40 magnification).

## Hormone secretion assays

Mouse islets were cultured overnight in RPMI (supplemented with 0.5 mg/mL BSA) then transferred to equilibration media (DMEM (no glucose) with 10% FBS, 0.5 mg/mL BSA, 10 mM HEPES, and 0.5 mM $CaCl_2$) supplemented with 5.6 mM glucose for 1 h at 37 °C, 5% $CO_2$. Islets were picked on ice into a 24-well plate (Corning) containing 400 μL of secretion media (DMEM (no glucose) with 0.5 mg/mL BSA, 10 mM HEPES, and 0.5 mM $CaCl_2$) supplemented with the glucose concentrations and treatments indicated in figure legends then cultured for 1 h at 37 °C, 5% $CO_2$. Secretion was halted by transferring the plate to ice for 10 min and supernatants were collected in low retention 1.6 mL centrifuge tubes. Supernatants were supplemented with 1:100 mammalian protease inhibitor cocktail and stored at −20 °C until analyzed. Secreted insulin was measured as per manufacturer instructions with ALPCO insulin chemiluminescence ELISA kits (15 islets per sample; catalog #: 80-INSHU-CH01) or Mercodia mouse insulin ELISA kits (50 islets per sample; catalog #: 10-1247-01); secreted SST was measured as per manufacturer instructions with Phoenix Pharmaceuticals SST chemiluminescent EIA kits (50 islets per sample; CEK-060-03).

## NKA immunoblotting

Mouse islets were isolated and incubated overnight in RPMI supplemented with 5.6 mM glucose at 37 °C and 5% $CO_2$. The islets were transferred to KRHB supplemented with treatments indicated in figure legends for 15 min at 25 °C, washed with phosphate-buffered saline supplemented with identical treatments along with 20 μL/mL Halt protease/phosphatase inhibitor cocktail (Halt PPI; catalog #: 78442), and frozen in a dry ice ethanol bath. Islets were lysed on ice in RIPA buffer supplemented with Halt PPI, then cell lysates were resolved on nitrocellulose membranes. Immunoblots were blocked for 1 h in Tris-buffered saline with 0.1% Tween 20 (TBST) supplemented with 5% BSA. All primary and secondary antibodies were diluted in TBST supplemented with 0.1% BSA. Immunoblots were probed with 1:500 rabbit anti-phospho-ATP1A1 (Y10) (p-ATP1A1$^{Y10}$; catalog #: PA5-17061; Thermo Fisher) followed by 1:2500 goat anti-rabbit HRP-conjugated secondary (catalog #: W4011; Promega, Madison, WI). p-ATP1A1$^{Y10}$ protein bands were visualized with SuperSignal™ West Pico Plus (SuperSignal Pico; catalog #: 34580; Thermo Fisher) utilizing a Bio-Rad Digital ChemiDoc MP (ChemiDoc). Immunoblots were stripped for 20 min with Restore™ Western Blot Stripping Buffer (catalog #: 21059; Thermo Fisher) and re-probed with 1:500 mouse anti-ATP1A1 followed by 1:2500 goat anti-mouse HRP-conjugated secondary (catalog #: W4021; Promega). ATP1A1 bands were visualized with SuperSignal Pico utilizing the ChemiDoc. Uncropped and unprocessed immunoblot scans are displayed in Supplementary Fig. 5.

## Data analysis

Islet $[Ca^{2+}]_i$, $[cAMP]_i$, $[Na^+]_i$, and NKA immunofluorescence were analyzed using Zeiss Zen software, Nikon Elements software, and the ImageJ Fiji image processing pack. Axon Clampfit software was utilized to quantify β-cell SST-induced currents and $V_m$ as well as perform cross-correlation analyses. All β-cell SST-induced currents are median values of 10 or more consecutive traces. Islet $[Ca^{2+}]_i$ plateau fraction was defined as the fraction of time during which islet $[Ca^{2+}]_i$ was ≥50% of islet $[Ca^{2+}]_i$ oscillation amplitude. Heatmaps were generated using the MATLAB imagesc function. Period analysis of $\delta G_i$ islet $[Ca^{2+}]_i$ oscillations was carried out utilizing the MATLAB detrend and findpeaks functions. Immunoblots were analyzed using Bio-Rad Image Lab 5.0. Figures were prepared utilizing Adobe Illustrator. Statistical analyses were carried out utilizing Microsoft Excel and GraphPad Prism 9.2.0 as indicated in figure legends; data were compared utilizing paired or unpaired two-sample t-tests, one-sample t-tests, or one-way analysis of variance (ANOVA) with Šidák's post-hoc multiple comparisons tests. Data were normalized when appropriate as indicated in figure legends. Unless stated otherwise, data are presented as mean values ± standard error (SEM) for the specified number of samples (n). Differences were considered significant for $P \leq 0.05$.

## Reporting summary

Further information on research design is available in the Nature Research Reporting Summary linked to this article.

## Data availability

The data that support this study are available from the corresponding authors upon reasonable request. Source data are provided with this paper.

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

## Acknowledgements

The authors thank Dr. Kevin D. Wickman (University of Minnesota) for providing mice with a loxP-flanked *Kcnj6* exon 4. We are also grateful for the confocal microscopy resources provided by the Vanderbilt Cell Imaging Shared Resource (CISR; supported by NIH grants CA68485, DK20593, DK58404, DK59637, and EY08126). This research was performed with the support of the Integrated Islet Distribution Program (https://iidp.coh.org/). We especially thank the organ donors and their families. Funding: These studies have been supported by a Vanderbilt Integrated Training in Engineering and Diabetes Grant (T32DK101003), an Initiative for Maximizing Student Development at Vanderbilt Grant (T32GM139800), a Multidisciplinary Training in Molecular Endocrinology Grant (T32DK007563), National Institutes of Health Grants (DK-097392 and DK-115620), an American Diabetes Association Grant (1-17-IBS-024), a Juvenile Diabetes Research Foundation Grant (2-SRA-2019-701-S-B), and a Pilot and Feasibility grant through the Vanderbilt Diabetes Research and Training Center Grant (P60-DK-20593).

## Author contributions

M.T.D. and D.A..J. formulated and designed experiments. M.T.D., P.K.D., K.E.Z., A.Y.N., C.M.S., J.R.D., N.M.W., J.C.L., C.F.S., and L.D.R. performed experiments. M.T.D. and D.A.J. analyzed data. M.T.D. and D.A.J. interpreted experimental results. M.T.D. and D.A.J. prepared figures. M.T.D. and D.A.J. drafted the manuscript. M.T.D., P.K.D., K.E.Z., A.Y.N., C.M.S., J.R.D., N.M.W., J.C.L., C.F.S., L.D.R., and D.A..J. approved the final manuscript submitted for publication.

## Competing interests

The authors declare no competing interests.
