## [Peer Review File · Nature Communications]

Gi/o protein-coupled receptor inhibition of beta-cell electrical excitability and insulin secretion depends on Na⁺/K⁺ ATPase activationReviewers' Comments:

Reviewer #1:

Remarks to the Author:

This is an important and exciting paper that identifies Na/K-ATPases (NKA) as the target of GPCR signaling in beta cells. This paper will certainly be an important contribution to the field. The authors demonstrate that, contrary to prior work, GIRK signaling is not the Gi/o target, as shown by GIRK2-floxed:Pdx1-Cre and Tertiapin-Q inhibition. NKAs are identified as the target of somatostatin (SST) signaling by ouabain as well as $0K^+$ in the bath solution, and the authors present beautiful high-resolution Vm, Ca, gK (K conductance), and cAMP measurements to support a model in which PKA signaling inhibits and Src kinases activate NKAs.

I have no major concerns about the rigor of the presented studies or the novel concept that SST signaling activates NKA, which is well supported by pharmacologic studies. However, the authors did not do a thorough job of showing physiological relevance of the mechanism they have identified. They do not determine whether endogenous islet SST works directly on beta cell Gi/o, or indirectly via the alpha cell, which would instead implicate the SST-dependent loss of beta cell GLP1r signaling. Fortunately, my concerns are easily addressed.

Major:

1. The title of the paper implies that SST works directly on the beta cell via Gi/o signaling. Physiologically, it may actually be that SST suppresses glucagon secretion, and beta cells experience the loss of glucagon signaling through the beta cell GLP1r/Gs/cAMP pathway. Indeed, in mouse islets SST is thought to have a strong inhibitory effect on alpha cell glucagon secretion (primarily via SSTR2), and a relatively weaker effect on beta cell insulin secretion (primarily via SSTR3). Viewed in this context, the clonidine and beta cell Gi/o DREADD experiments in Fig. 4A-F do not actually address the physiological mechanism of SST action, despite the claim on lines 233-234.

Testing whether blocking SST effect on the alpha (e.g. with SSTR2 antagonists) or beta cell (e.g. with SSTR3 antagonists) would identify the relevant beta signaling pathway – Gi/o vs. Gs – under moderate glucose stimulation (9G). The authors should bear in mind that the presence of physiological amino acids (listed in the Supplement of Lai and Gilon, Diabetes 2018; PMID: 30115649) could change the alpha cell input. As it stands, the authors biased the experiments to exclude the alpha cell by using high glucose without amino acids present.

2. The DREADD experiments in Fig. 4G are important to demonstrate that SST regulates beta cells under physiological conditions. Notably, the calcium measurements in Fig 4G are missing the no-CNO controls. Second, the authors report the period of oscillations and the AUC but not the plateau fraction. Does delta cell blockade increase the plateau fraction at 9G? Does SST supplementation reduce it?

3. Clearly NKA is essential for oscillations, which are terminated by ouabain or $0K$. Can low-dose ouabain be used to test the effect of moderate NKA inhibition on Ca plateau fraction? Is there a dose-dependent effect?

Minor

1. Methods: How was calcium plateau fraction quantified? This is missing.

2. The authors never say the SST concentration they used on mouse islets – this should be listed in the Figure 1A legend and associated text.

3. Line 169: The authors claim calcium oscillations ceased in 96.1 percent of islets after removal of

extracellular K. How could this be determined? It appears that in Fig. 2B the 0 K solution wasn't present long enough to quantify loss of oscillations.

4. Line 138, remove "add some references".

5. Line 196: remove "suggesting that NKA regulation of beta cell calcium handling is not restricted to Gi/o signaling". this cannot be concluded from the a cross correlation analysis.

6. Line 234: "these data confirm that direct stimulation of beta cell Gi/o signaling initiates islet calcium oscillation". This language is too imprecise - Gi/o signaling cannot initiate calcium oscillations.

7. Line 321: the "area" of islets was not measured. AUC?

Reviewer #2:

Remarks to the Author:

The study by Dickerson et al. investigated the mechanism of Gi-mediated inhibition of beta cell electrical activity and Ca²⁺ oscillation, which are critical for glucose stimulated insulin secretion. The authors reached the conclusion that the main target of somatostatin receptor activation (and subsequent Gi protein) is the sodium/potassium pump (NKA). The manuscript was constructed in a logical order with a large amount of data. However, I feel the conclusion should be supported by more unequivocal data. I have a few comments as follows.

1. The current recordings demonstrated that sst increased membrane conductance (which can be calculated using the linear currents elicited by voltage ramp), an effect that was reduced by abolishing GIRK2 expression or TPQ. This indicates that GIRK plays a role in sst-induced membrane repolarisation. The data presented also highlighted a component that is not GIRK mediated and authors argued that was other than a K⁺ conductance. But from Fig. 1B, E and N, it can be seen that currents reversed at -70 mV when SST was present. This argues against the authors' conclusion and suggests SST activated a K⁺ conductance. The fact that diazoxide inhibited Ca²⁺ oscillation when extracellular K⁺ was excluded does not support an alternative conductance – this manoeuvre would shift the K⁺ reversal potential to even more negative membrane potential and naturally when KATP channels are active, the cell would repolarise. Actually, this was demonstrated in Fig. 2H, where a transient repolarisation was observed when K⁺ was removed, suggesting a shift in reversal potentials.
2. The KCNJ6 knockout is a good approach, but I am wondering if the other GIRK channels may also contribute sst-induced repolarisation. For example, in human islets, there are at least 3 different GIRK channels with KCNJ5 having the highest expression. It is unclear that the sst-induced Ca²⁺ oscillations observed in human islets can be attributed to NKA.
3. I also found the effect of SST is rather slow. It took >3 mins to induce membrane repolarisation. This is quite different from what has been published before (<1min, in the presence of tolbutamide). Could the authors please explain the discrepancy? Also, it would be good if the authors can show the membrane potential recordings made in GIRK2 KO islets. The Ca²⁺ traces are noisier in GIRK2 KO islets – could this be because the magnitude of repolarisation is different?
4. It will be strong support for their conclusion if the authors can actually measure NKA currents in the presence and absence of SST. This can be done by altering extracellular K⁺ concentration acutely.
5. The authors suggested that the slow SST-induced Ca²⁺ oscillation is due to an interplay between Ca²⁺ and cAMP: 'Gi/o-GPCR signalling (a) activates β -cell NKAs leading to inhibition of Ca²⁺ influx; (b) this drives cAMP generation that inhibits NKA activity and stimulates Ca²⁺ entry; (c) this in turn, reduces [cAMP]_i and PKA inhibition of NKA ...'. How does this reconcile with the Gi protein's effect on reducing cAMP production? If the Gi activation persisted throughout the recording, it is likely that cAMP would be low – and less oscillatory. The alternative explanation could be that the oscillation was due to SSTR desensitisation and recycling of the receptors. This possibility should be explored (at least discussed).

6. The forskolin effect cannot be only attributed to NKA. Forskolin/cAMP/PKA can also inhibit GIRK channels (PMID: 23211299), which can contribute to the reduced SST effect. In fact, in the presence of forskolin, ramp-triggered current did not reverse at -70 mV, which may suggest SST-activated K⁺ conductance was abolished by the adenylyl cyclase activator.

7. Authors suggested NKA activity is behind the Ca²⁺ oscillation by demonstrating beta cell Na⁺ oscillation coincided with Ca²⁺ oscillation. Can this be simply due to electrical activity and opening of voltage-gated Na⁺ channels? Although beta cell Nav channels contribute little to its action potential firing, when cells are at a slightly more repolarised membrane potential (which can be the case since sst repolarises beta cells) its involvement in electrical activity generation can be much higher.

REVIEWER COMMENTS

Reviewer #1 (Remarks to the Author):

This is an important and exciting paper that identifies Na/K-ATPases (NKA) as the target of GPCR signaling in beta cells. This paper will certainly be an important contribution to the field. The authors demonstrate that, contrary to prior work, GIRK signaling is not the Gi/o target, as shown by GIRK2 floxed:Pdx1-Cre and Tertiapin-Q inhibition. NKAs are identified as the target of somatostatin (SST) signaling by ouabain as well as OK+ in the bath solution, and the authors present beautiful high-resolution Vm, Ca, gK (K conductance), and cAMP measurements to support a model in which PKA signaling inhibits and Src kinases activate NKAs.

I have no major concerns about the rigor of the presented studies or the novel concept that SST signaling activates NKA, which is well supported by pharmacologic studies. However, the authors did not do a thorough job of showing physiological relevance of the mechanism they have identified. They do not determine whether endogenous islet SST works directly on beta cell Gi/o, or indirectly via the alpha cell, which would instead implicate the SST-dependent loss of beta cell GLP1r signaling. Fortunately, my concerns are easily addressed.

We would like to thank the Reviewer for their time in reviewing our manuscript. We appreciate the helpful feedback and assistance in ensuring the highest quality publication possible. We have thoroughly addressed all of the reviewer's comments and revised the manuscript accordingly. These changes have strengthened the manuscript and are summarized below. We believe that our manuscript is greatly improved and that it is suitable for publication in *Nature Communications*.

Major:

1. The title of the paper implies that SST works directly on the beta cell via Gi/o signaling. Physiologically, it may actually be that SST suppresses glucagon secretion, and beta cells experience the loss of glucagon signaling through the beta cell GLP1r/Gs/cAMP pathway. Indeed, in mouse islets SST is thought to have a strong inhibitory effect on alpha cell glucagon secretion (primarily via SSTR2), and a relatively weaker effect on beta cell insulin secretion (primarily via SSTR3). Viewed in this context, the clonidine and beta cell Gi/o DREADD experiments in Fig. 4A-F do not actually address the physiological mechanism of SST action, despite the claim on lines 233-234.

Testing whether blocking SST effect on the alpha (e.g. with SSTR2 antagonists) or beta cell (e.g. with SSTR3 antagonists) would identify the relevant beta signaling pathway – Gi/o vs. Gs – under moderate glucose stimulation (9G). The authors should bear in mind that the presence of physiological amino acids (listed in the Supplement of Lai and Gilon, Diabetes 2018; PMID: 30115649) could change the alpha cell input. As it stands, the authors biased the experiments to exclude the alpha cell by using high glucose without amino acids present.

The reviewer raises an excellent point and we have added the following experiments to the manuscript in order to address this: **(1)** Islet GLP1Rs were pharmacologically inhibited with Exendin-3 (9-39) prior to activation of SST-induced Ca²⁺ oscillations (Fig. 4G-4I). Inhibition of islet GLP1R signaling did not prevent SST-induced Ca²⁺ oscillations; this suggests that stimulation of β-cell Gi/o signaling rather than reduced glucagon signaling mediates the effect of SST (Fig. 4G-4I). **(2)** This was confirmed by another new data set showing that activation of α-cell SSTR2s with the selective SSTR2 agonist L-054,264 does not initiate SST-induced islet Ca²⁺ oscillations (Fig. 4J-4L). Taken together these findings strongly suggest that SST-induced alterations in GCG secretion do not cause islet Ca²⁺ oscillations under depolarized conditions. This

evidence supports a mechanism by which β -cell SSTR signaling activates NKA activity that controls islet Ca^{2+} oscillation frequency.

2. The DREADD experiments in Fig. 4G are important to demonstrate that SST regulates beta cells under physiological conditions. Notably, the calcium measurements in Fig 4G are missing the no-CNO controls. Second, the authors report the period of oscillations and the AUC but not the plateau fraction. Does delta cell blockade increase the plateau fraction at 9G? Does SST supplementation reduce it?

We appreciate these helpful suggestions and have added the following to the manuscript: A no-CNO control $\delta\text{G}_{i/o}$ -DREADD islet $[\text{Ca}^{2+}]_i$ recording was added to Fig. 4M along with analysis of this data (Fig. S3) as well as islet Ca^{2+} plateau fraction (Fig. 4P). This analysis determined that while $\delta\text{G}_{i/o}$ -DREADD activation increased islet Ca^{2+} oscillation frequency, it had no effect on islet Ca^{2+} plateau fraction (Fig. 4N and 4P). However, addition of exogenous SST not only increased the period of islet Ca^{2+} oscillations by $74.1 \pm 20.4\%$ (Fig. 4N: $P < 0.001$), but also reduced plateau fraction by $18.6 \pm 1.2\%$ (Fig. 4P; $P < 0.05$).

3. Clearly NKA is essential for oscillations, which are terminated by ouabain or 0K^+ . Can low-dose ouabain be used to test the effect of moderate NKA inhibition on Ca plateau fraction? Is there a dose-dependent effect?

The reviewer poses an interesting question regarding the dose-dependence of ouabain inhibition of NKAs. Ouabain affinity varies greatly depending on NKA α subunit isoform; NKA $\alpha 1$, the isoform expressed in β -cells, displays very low (1×10^{-4} M) ouabain sensitivity (*Life Sci.*, **1978**, 23:2735–2744). There is also evidence that low levels of ouabain can paradoxically stimulate NKA-mediated Na^+/K^+ exchange as well as cell signaling (*Biochim Biophys Acta.*, **2016** Nov; 1863(11): 2624–2636). As these factors complicate the issue, we made the decision to restrict use of ouabain to doses high enough to block NKA-mediated Ca^{2+} oscillations. However, in large islets we were unable to completely inhibit NKA activity with the Src tyrosine kinase inhibitor dasatinib (Fig. 7A and 7B). This data suggests that partial NKA inhibition results in acceleration of islet Ca^{2+} oscillations and increased Ca^{2+} plateau fraction. We plan to follow up on this in future genetic-based experiments testing how reduced β -cell NKA expression and/or surface localization affects islet Ca^{2+} handling and insulin secretion, which will be included in a forthcoming publication.

Minor

1. Methods: How was calcium plateau fraction quantified? This is missing.

Islet Ca^{2+} plateau fraction was defined as the percentage of time in which islet Ca^{2+} (based on indicator fluorescence) was $\geq 50\%$ of Ca^{2+} oscillation amplitude. In the majority of cases where there was little to no change in F_{max} or F_{min} over time Ca^{2+} oscillation amplitude was calculated from the following equation. In some cases, F_{min} and/or F_{max} changed substantially over time (i.e. during $\delta\text{G}_{i/o}$ -DREADD experiments). Thus, the above approach was utilized with modifications. Briefly, F_{min} and F_{max} were calculated over five minute intervals surrounding each data point. In this manner, a rolling Ca^{2+} plateau threshold was determined that more accurately represented Ca^{2+} plateau fraction.

$$\text{Ca}^{2+} \text{ plateau threshold} = 0.5 * (F_{\text{max,global}} - F_{\text{min,global}}) + F_{\text{min,global}}$$

$$\text{Ca}^{2+} \text{ plateau threshold} = 0.5 * (F_{\text{max, 300 Sec}} - F_{\text{min, 300 Sec}}) + F_{\text{min, 300 Sec}}$$

2. The authors never say the SST concentration they used on mouse islets – this should be listed in the Figure 1A legend and associated text.

We thank the reviewer for pointing out this mistake. The manuscript has been revised to indicate the concentration of SST used in figure legends and throughout the paper as appropriate.

3. Line 169: The authors claim calcium oscillations ceased in 96.1 percent of islets after removal of extracellular K. How could this be determined? It appears that in Fig. 2B the 0 K solution wasn't present long enough to quantify loss of oscillations.

We thank the reviewer for this observation and we have modified the text accordingly to address this concern. The text now reads as follows:

“ $[\text{Ca}^{2+}]_i$ remained elevated in $97.1 \pm 2.9\%$ of islets following NKA inhibition with ouabain ($150 \mu\text{M}$; $P < 0.001$) and in $96.1 \pm 1.9\%$ of islets after removal of extracellular K^+ (Fig. 2A-2C; $P < 0.0001$).”

4. Line 138, remove “add some references”.

We thank the reviewer for finding this mistake. The manuscript has been revised to address this issue.

5. Line 196: remove “suggesting that NKA regulation of beta cell calcium handling is not restricted to Gi/o signaling”. this cannot be concluded from the cross correlation analysis.

We agree and have removed this passage from the manuscript.

6. Line 234: “these data confirm that direct stimulation of beta cell Gi/o signaling initiates islet calcium oscillation”. This language is too imprecise - Gi/o signaling cannot initiate calcium oscillations.

This sentence in the manuscript has been revised to address the reviewer's comment. The text now reads

“These data confirm that under depolarizing conditions direct stimulation of β -cell $\text{G}_{i/o}$ signaling transiently hyperpolarizes V_m by activating NKAs.”

7. Line 321: the “area” of islets was not measured. AUC?

We apologize that this point was not clear and have addressed it as follows: because we observed bifurcated responses to dasatinib (Dasa) and NSC 87877 (NSC), we quantified relative islet area for these particular Ca^{2+} experiments and found a correlation between islet size and inhibition of SST-induced islet Ca^{2+} oscillations. To make this clear we have added new panels to Fig. (7D and 7H) showing quantification of relative islet areas.

Reviewer #2 (Remarks to the Author):

The study by Dickerson et al. investigated the mechanism of Gi-mediated inhibition of beta cell electrical activity and Ca²⁺ oscillation, which are critical for glucose stimulated insulin secretion. The authors reached the conclusion that the main target of somatostatin receptor activation (and subsequent Gi protein) is the sodium/potassium pump (NKA). The manuscript was constructed in a logical order with a large amount of data. However, I feel the conclusion should be supported by more unequivocal data. I have a few comments as follows.

We would like to thank the reviewer for the constructive feedback, which has substantially strengthened our manuscript. We have thoroughly addressed all of the reviewer's comments and revised the manuscript accordingly in order to ensure the highest quality publication possible. We believe that our manuscript is greatly improved and that it is suitable for publication in *Nature Communications*. These changes are summarized below.

1. The current recordings demonstrated that SST increased membrane conductance (which can be calculated using the linear currents elicited by voltage ramp), an effect that was reduced by abolishing GIRK2 expression or TPQ. This indicates that GIRK plays a role in SST-induced membrane repolarization. The data presented also highlighted a component that is not GIRK mediated and authors argued that was other than a K⁺ conductance. But from Fig. 1B, E and N, it can be seen that currents reversed at -70 mV when SST was present. This argues against the authors' conclusion and suggests SST activated a K⁺ conductance. The fact that diazoxide inhibited Ca²⁺ oscillation when extracellular K⁺ was excluded does not support an alternative conductance - this manoeuvre would shift the K⁺ reversal potential to even more negative membrane potential and naturally when KATP channels are active, the cell would repolarize. Actually, this was demonstrated in Fig. 2H, where a transient repolarization was observed when K⁺ was removed, suggesting a shift in reversal potentials.

We thank the reviewer for raising these important points and our response is as follows: the current recordings shown in Fig. 1B, 1E, and 1N contain background currents, which include K⁺ currents that indeed reverse at approximately -70 mV. However, when the background currents present before SST treatment are subtracted from those after, what remains are SST-induced currents (Fig. 1C, 1F, 1O; also see below). SST-induced currents did not reverse within the range of voltages we examined (-100 to -50mV) and when plotted together there was no difference between WT and GIRK2 KO β -cell SST-induced currents (see left trace below). Interestingly, while the difference was not significant, SST-induced currents in WT β -cells treated with tertiapin-Q (TPQ) trended lower than untreated controls (see right trace below). Furthermore, while TPQ did not terminate SST-induced Ca²⁺ oscillations there was a small but significant increase in Ca²⁺ oscillation frequency (Fig. 1S). This may suggest a role for a non-GIRK2 GIRK channel in hyperpolarizing β -cell V_m. These observations have now been emphasized in the manuscript.

The reviewer is also correct that omission of extracellular K⁺ would be expected to left-shift the reversal potential of K⁺, which was in fact our original goal. We reasoned that this approach would assist us in measuring small GIRK currents by magnifying the chemical driving force of K⁺ through these channels; however, to our surprise

removal of extracellular K^+ actually inhibited SST-induced β -cell currents. To reinforce this point, we have now included episodic β -cell currents recorded from intact mouse islets held at -80mV demonstrating that SST activates an outward current that is inhibited following removal of K^+ from the recording solution (Fig. 2H and 2I). Importantly, as the reviewer points out, removal of extracellular K^+ would left-shift the equilibrium potential (E_K) of K^+ and increase K^+ channel activity at -80mV. However, we instead observed inhibition of outward SST-induced currents, which strongly suggests that this effect is not simply due to a shift in E_K .

We were also concerned that V_m depolarization resulting from NKA inhibition (0 K^+ or ouabain) could be so great that it prevents K^+ channel-mediated V_m hyperpolarization. Thus, we added diazoxide at the end of Ca^{2+} imaging experiments shown in Fig. 2B to provide evidence that K^+ channel activation can still hyperpolarize V_m . However, the reviewer is correct that this does not establish that a non- K^+ channel conductance mediates β -cell SST-induced V_m hyperpolarization. Thus, we have revised the manuscript as proposed by the reviewer, which no longer indicates that this data supports an alternative conductance.

2. The KCNJ6 knockout is a good approach, but I am wondering if the other GIRK channels may also contribute sst-induced repolarization. For example, in human islets, there are at least 3 different GIRK channels with KCNJ5 having the highest expression. It is unclear that the sst-induced Ca^{2+} oscillations observed in human islets can be attributed to NKA.

This is an important observation that we address as follows: single mouse islet cell RNA sequencing strongly suggests that mouse β -cells predominantly express *Kcnj6* (gene encoding GIRK2) (*Mol Metab*, **2017**, 5:449-458). Thus, we believe that a combined approach of genetic knockout and pharmacological inhibition is appropriate to interrogate the role of GIRK2 in $G_{i/o}$ -GPCR-mediated mouse β -cell hyperpolarization. The reviewer makes a good point that human islets express transcripts encoding GIRK1 (*KCNJ3*), GIRK2 (*KCNJ6*), and GIRK4 (*KCNK5*) channels (*Am J Physiol Endocrinol Metab*, **2012**, Nov 1; 303(9): E1107–E1116). Therefore, to determine if human β -cell GIRK channels contribute to $G_{i/o}$ -GPCR-mediated V_m hyperpolarization, we have now included data with pharmacological inhibition of GIRK channels with TPQ, which has previously been shown to have effects on human β -cells (*Am J Physiol Endocrinol Metab*, **2012**, Nov 1; 303(9): E1107–E1116). This new study utilizing TPQ found significant SST-induced decreases in human islet Ca^{2+} even with GIRK channel inhibition (Fig. 5G and 5H). These experiments strongly suggest that GIRK channels are not the sole mediator of SST-induced β -cell V_m hyperpolarization. However, we have modified the manuscript (see revised results and discussion) to include possible contributions of GIRK channels to SST regulation of human β -cell function.

3. I also found the effect of SST is rather slow. It took >3 mins to induce membrane repolarisation. This is quite different from what has been published before (<1min, in the presence of tolbutamide). Could the authors please explain the discrepancy? Also, it would be good if the authors can show the membrane potential recordings made in GIRK2 KO islets. The Ca^{2+} traces are noisier in GIRK2 KO islets – could this be because the magnitude of repolarisation is different?

We thank the reviewer for this important observation. The time required for SST-induced V_m hyperpolarization was indeed longer than previously observed (*Am J Physiol Endocrinol Metab*, **2012**, Nov 1; 303(9): E1107–E1116); however, we commonly observed more rapid decreases in β -cell action potential firing that preceded V_m hyperpolarization. Furthermore, there were key differences in the experimental conditions employed. **(1)** V_m recordings in Kailey et al. were performed in dispersed human β -cells whereas in this manuscript we utilized intact human islets. Thus, it is probable that this delay was due in-part to the time required for SST to diffuse throughout intact islets. **(2)** V_m recordings in Kailey et

al. were conducted in the presence of 6mM glucose while here we employed 20mM glucose. A shift from 6 to 20 mM glucose would increase β -cell cAMP levels (*Ups J Med Sci*, **2012**, 117(4): 355–369), which we have shown decreases NKA activity in a PKA-dependent manner and thus could contribute to the observed lag time. The reviewer also makes a good suggestion and we have now included a GIRK2 KO islet V_m recording (Fig. S1). Although the reviewer noted that the representative Fura-2 AM GIRK2 KO Ca^{2+} recording is a bit noisier than the WT control shown, we did not observe significant differences in SST-mediated V_m repolarization between WT and GIRK2 KO β -cells (Fig. 1D and 1G).

4. It will be strong support for their conclusion if the authors can actually measure NKA currents in the presence and absence of SST. This can be done by altering extracellular K^+ concentration acutely.

We thank the reviewer for suggesting this critical experiment. We have now included episodic β -cell currents recorded in intact mouse islets held at -80mV showing SST-induced outward currents, which are subsequently inhibited by removal of K^+ from the recording solution (Fig. 2H and 2I). This experiment substantiates that NKAs can be activated by SST.

5. The authors suggested that the slow SST-induced Ca^{2+} oscillation is due to an interplay between Ca^{2+} and cAMP: ' $G_{i/o}$ -GPCR signaling (a) activates β -cell NKAs leading to inhibition of Ca^{2+} influx; (b) this drives cAMP generation that inhibits NKA activity and stimulates Ca^{2+} entry; (c) this in turn, reduces [cAMP]; and PKA inhibition of NKA ...'. How does this reconcile with the G_i protein's effect on reducing cAMP production? If the G_i activation persisted throughout the recording, it is likely that cAMP would be low – and less oscillatory. The alternative explanation could be that the oscillation was due to SSTR desensitization and recycling of the receptors. This possibility should be explored (at least discussed).

The reviewer highlights several important observations and is correct that SSTR desensitization could be an alternative explanation for the oscillatory effects that were observed during islet SST treatment. Therefore, we have now included details regarding receptor desensitization in our discussion. However, our data suggest that SSTR-mediated reductions in adenylyl cyclase (AC) activity may have a smaller effect on β -cell function than previously suggested, as we summarize here.

Utilizing a genetically encoded cAMP indicator (cAMP_{Pr}) we observed a large decrease in cAMP following tolbutamide-mediated K_{ATP} inhibition and resulting Ca^{2+} influx, which we hypothesize is due to Ca^{2+} -mediated inhibition of ACs and/or activation of phosphodiesterases (PDEs). This suggests that Ca^{2+} plays a critical role in controlling β -cell cAMP, as has been previously reported (such as in Tenner et al. *eLife* **2020**;9:e55013). Indeed, β -cell cAMP increased during NKA-mediated decreases in Ca^{2+} , pointing to either **(1)** SSTR desensitization/turnover or **(2)** alleviation of Ca^{2+} -dependent AC inhibition and/or decreased Ca^{2+} -mediated PDE activation, which overrode SSTR-mediated AC inhibition. To test this, we clamped Ca^{2+} at a low level with diazoxide and measured islet Na^+ as an indicator of NKA activity. Under these conditions, SST induced sustained non-oscillatory decreases in islet Na^+ , which could be inhibited with conditions that increase islet cAMP (Fig. 6E and 6H). Taken together, these data suggest that islet SSTR activity is maintained during periods of prolonged stimulation and that Ca^{2+} may be a more potent regulator of β -cell cAMP production than SSTR-mediated AC inhibition. Moreover, inhibition of PKA activity resulted in non-oscillatory NKA activity and a continual SST-induced reduction of β -cell Ca^{2+} (Fig. 6A–6C) and Na^+ (Fig. 6F and 6I). While these data suggest that SSTR activity may be constant in β -cells in the presence of SST, we cannot rule out that SSTR desensitization plays an important role in the process. This is now detailed in the discussion section of the manuscript.

6. The forskolin effect cannot be only attributed to NKA. Forskolin/cAMP/PKA can also inhibit GIRK channels (PMID: 23211299), which can contribute to the reduced SST effect. In fact, in the presence of

forskolin, ramp-triggered current did not reverse at -70 mV, which may suggest SST-activated K⁺ conductance was abolished by the adenylyl cyclase activator.

As the reviewer points out, forskolin-mediated increases in cAMP inhibited SST-induced outward currents, which we attribute to NKA as these currents were present in GIRK2 KO β -cells and when GIRK channels were inhibited with TPQ. Furthermore, we clamped islet V_m at a hyperpolarized potential by activating K_{ATP} with diazoxide and investigated Na^+ handling (with an ING-2 Na^+ dye). Using this approach, we demonstrated that SST induces Na^+ efflux, which was inhibited by increases in islet cAMP (activation of G_s -coupled GLP1Rs with liraglutide). This strongly suggests that β -cell NKAs are activated by SST to a great enough extent to alter $[Na^+]_i$ concentrations. Moreover, physiological elevations in β -cell cAMP inhibit NKA activity. However, the reviewer is correct in that forskolin/liraglutide could impact K^+ channel activity. Thus, we have now acknowledged this possibility in the manuscript.

7. Authors suggested NKA activity is behind the Ca^{2+} oscillation by demonstrating beta cell Na^+ oscillation coincided with Ca^{2+} oscillation. Can this be simply due to electrical activity and opening of voltage-gated Na^+ channels? Although β -cell Nav channels contribute little to its action potential firing, when cells are at a slightly more repolarised membrane potential (which can be the case since sst repolarises beta cells) its involvement in electrical activity generation can be much higher.

The reviewer poses an interesting question regarding possible roles for Na^+ permeable ion channels in controlling the islet $[Na^+]_i$ oscillations that accompany $[Ca^{2+}]_i$ oscillations. Mouse β -cell voltage-dependent Na^+ (Nav) channels are inactive at voltages above -50mV (*J. Physiol*, **2014**, 592.21:4677–4696) and as the resting V_m of islets stimulated with SST (\sim 77mV; Fig. 1D) was not substantially more hyperpolarized than expected for islets under low glucose conditions (\sim 73mV; *Diabetes*, **2015** Nov; 64(11): 3818–3828), it is unlikely that Nav channels are significantly more active during SST treatment. For these reasons, Nav channels would not be predicted to account for sustained (5-10 minutes) SST-mediated increases in islet Na^+ that we observed (Fig. 3A). This does not rule out the potential of other β -cell Na^+ channels that are activated by Ca^{2+} such as TRPM4 and TRPM5. Therefore, to eliminate contributions from Na^+ conducting channels, a fluorescent Na^+ indicator was utilized to measure SST-induced changes in islet Na^+ under hyperpolarizing conditions (diazoxide) where Nav channels are inactive and intracellular Ca^{2+} is low (Fig 6D-6I). Ouabain-sensitive Na^+ efflux under these conditions demonstrates that SSTR signaling stimulates β -cell NKA activity (Fig. 6D and 6G). However, we have added discussion of potential contributions of other Na^+ channels such as TRPM4 and TRPM5 to SST-induced islet $[Na^+]_i$ oscillations.

Reviewers' Comments:

Reviewer #1:

Remarks to the Author:

This paper is an outstanding contribution to the field. The authors have addressed my major comments very thoroughly, resulting in a much improved manuscript.

Minor

--These statements are misleading, and should be rewritten for clarity:

In the abstract, "Gi/o coupled somatostatin or α_2 adrenergic signaling induced oscillations in beta cell NKA activity, resulting in islet calcium fluctuations." Later in the abstract, "Gi/o-GPCR-mediated oscillations..." And in the Discussion, "NKAs, which control the frequency of islet calcium oscillations and initiate calcium oscillations independent of Katp channel activity".

Gi/o signaling cannot "induce" oscillations in NKA activity, nor can it "initiate calcium oscillations"; metabolism plays the role of generating oscillations. I think the main point the authors are trying to make is that Gi/o signaling induces NKA activity, which hyperpolarizes the beta cell, slows calcium oscillations, and reduces insulin secretion. Also, while the NKA-dependent hyperpolarization is independent of KATP channel activation, the oscillations are strongly KATP dependent under physiological circumstances - which is not what is written. Rewriting these three sentences to emphasize the big picture will help a lot of readers who are not experts in electrophysiology.

--The cartoon in Fig. 8 should be modified. It should have [cAMP]_i listed only once; it should have [cAMP]_i produced FROM adenylylase and have an activating arrow to PKA. Furthermore, there is no evidence that SST completely terminates calcium influx and insulin secretion. Thus, there should be an inhibitory arrow from NKA to the calcium channel, and an activating arrow from calcium to insulin granules.

--Unless it was done by hand, the software used to calculate plateau fraction should be indicated in the Methods.

Reviewer #2:

Remarks to the Author:

The authors have thoroughly addressed my comments with additional experiments and a more detailed discussion. I have only one minor comment: it would be great if the authors could include representative traces of the ramp recordings before and after the addition of somatostatin - as shown in Kailey et al. This will help readers to have a better understanding of the data.

Reviewer #1 (Remarks to the Author):

This paper is an outstanding contribution to the field. The authors have addressed my major comments very thoroughly, resulting in a much improved manuscript.

We thank the reviewer for these helpful suggestions and have edited the manuscript as indicated below to address each point. These changes have strengthened the manuscript and we believe that it is now suitable for publication in *Nature Communications*.

Minor

--These statements are misleading, and should be rewritten for clarity:

In the abstract, "Gi/o coupled somatostatin or $\alpha 2$ adrenergic signaling induced oscillations in beta cell NKA activity, resulting in islet calcium fluctuations." Later in the abstract, "Gi/o-GPCR-mediated oscillations..." And in the Discussion, "NKAs, which control the frequency of islet calcium oscillations and initiate calcium oscillations independent of Katp channel activity".

Gi/o signaling cannot "induce" oscillations in NKA activity, nor can it "initiate calcium oscillations"; metabolism plays the role of generating oscillations. I think the main point the authors are trying to make is that Gi/o signaling induces NKA activity, which hyperpolarizes the beta cell, slows calcium oscillations, and reduces insulin secretion. Also, while the NKA-dependent hyperpolarization is independent of KATP channel activation, the oscillations are strongly KATP dependent under physiological circumstances - which is not what is written. Rewriting these three sentences to emphasize the big picture will help a lot of readers who are not experts in electrophysiology.

We thank the reviewer for these helpful suggestions and have changed the statements accordingly. The modified sentences are as follows:

" $G_{i/o}$ -coupled somatostatin or $\alpha 2$ -adrenergic receptor activation stimulated β -cell NKA activity, resulting in islet Ca^{2+} fluctuations."

" β -cell membrane potential hyperpolarization resulting from $G_{i/o}$ -GPCR activation was dependent on NKA phosphorylation by Src tyrosine kinases."

"We found that $G_{i/o}$ -GPCR activation stimulates β -cell NKAs, which hyperpolarizes V_m independently of K_{ATP} channel activity and leads to decreased islet $[Ca^{2+}]_i$ oscillation frequency."

--The cartoon in Fig. 8 should be modified. It should have $[cAMP]_i$ listed only once; it should have $[cAMP]_i$ produced FROM adenylate cyclase and have an activating arrow to PKA. Furthermore, there is no evidence that SST completely terminates calcium influx and insulin secretion. Thus, there should be an inhibitory arrow from NKA to the calcium channel, and an activating arrow from calcium to insulin granules.

We thank the reviewer for this suggestion and have now altered Fig. 8 to include these changes.

--Unless it was done by hand, the software used to calculate plateau fraction should be indicated in the Methods.

Islet Ca²⁺ plateau fractions were calculated by hand utilizing Microsoft Excel; this was done in a blinded manner where treatment conditions were unknown before analysis.

Reviewer #2 (Remarks to the Author):

The authors have thoroughly addressed my comments with additional experiments and a more detailed discussion.

We thank the reviewer for their suggestions and feedback in strengthening our manuscript. We have addressed the reviewer's point as indicated below and we believe that our manuscript is now suitable for publication in *Nature Communications*.

I have only one minor comment: it would be great if the authors could include representative traces of the ramp recordings before and after the addition of somatostatin - as shown in Kailey et al. This will help readers to have a better understanding of the data.

We thank the reviewer for this suggestion and have now included representative β -cell voltage ramp traces before and after addition of SST as a Supplementary Fig. 4.